# Split Conformal Classification with Unsupervised Calibration

**Santiago Mazuelas**
BCAM-Basque Center for Applied Mathematics and
IKERBASQUE-Basque Foundation for Science
Bilbao, Spain
smazuelas@bcamath.org

## Abstract

Methods for split conformal prediction leverage calibration samples to transform any prediction rule into a set-prediction rule that complies with a target coverage probability. Existing methods provide remarkably strong performance guarantees with minimal computational costs. However, they require the use calibration samples composed by labeled examples different to those used for training. This requirement can be highly inconvenient, as it prevents the use of all labeled examples for training and may require acquiring additional labels solely for calibration. This paper presents an effective methodology for split conformal prediction with unsupervised calibration for classification tasks. In the proposed approach, set-prediction rules are obtained using unsupervised calibration samples together with supervised training samples previously used to learn the classification rule. Theoretical and experimental results show that the presented methods can achieve performance comparable to that with supervised calibration, at the expenses of a moderate degradation in performance guarantees and computational efficiency.

## 1 Introduction

Conformal prediction methods can significantly increase the reliability of machine learning techniques by providing prediction sets that contain the true response (class) with a coverage probability that can be chosen in advance (see e.g., [1–3]). In particular, methods for split (aka inductive) conformal prediction [2, 4] are particularly convenient since they can transform any prediction rule (possibly highly unreliable) into a set-prediction rule that complies with a target coverage probability. In these methods, the set-prediction rule is obtained using a fresh set of examples referred to as calibration samples. Existing methods for split conformal prediction provide remarkably strong performance guarantees with minimal computational costs. For general data distributions, they ensure a small coverage gap between the actual and target coverage probabilities. In addition, these methods only require to compute an empirical quantile from the calibration samples, which can be carried out with quasilinear complexity and negligible running time in practice. While conformal prediction has been developed for both classification and regression tasks, this paper focuses exclusively on classification problems, in line with other works for conformal prediction using weakly-supervised data [5, 6].

Conventional methods for split conformal prediction require the use supervised calibration samples composed by accurately labeled examples. This requirement forces to either leave aside a portion of labeled examples during the training phase or to acquire new labels solely for calibration. Both cases are problematic, the use of fewer samples for training can increase classification error, and the acquisition of new labeled examples is often costly. It is worth highlighting that obtaining new labels after the training phase may be even unfeasible in certain practical scenarios, including cases

39th Conference on Neural Information Processing Systems (NeurIPS 2025).

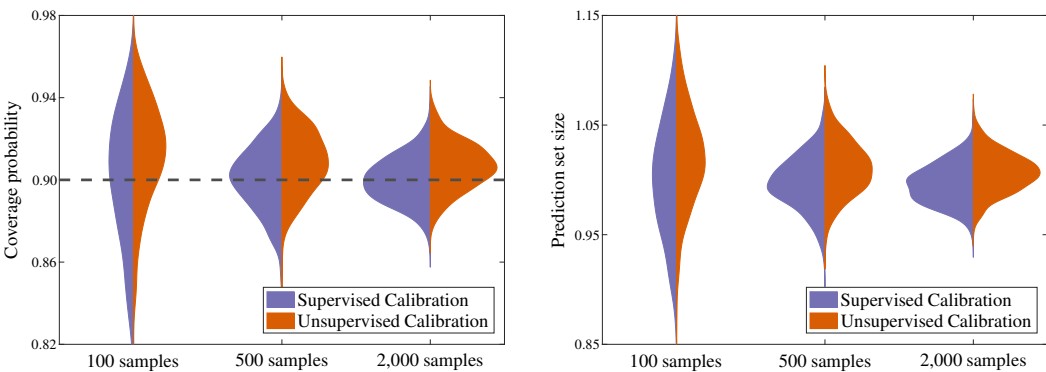

Figure 1: Coverage probabilities and prediction set sizes over 400 random partitions of 'CIFAR10' dataset with target coverage $1 - \alpha = 0.9$ and number of calibration samples ranging from 100 to $2,000$. The presented methods with unsupervised calibration achieve performance that is only slightly inferior to that of conventional methods with supervised calibration (coverage probabilities slightly less centered around 0.9). With both approaches, the coverage gap between the actual and target coverage probabilities decreases with more calibration samples and remains below 0.02 with high probability using a few thousand calibration samples.

where labels are protected by privacy regulations or correspond to future events that are not yet observable. Several methods have been recently proposed for situations with weakly-supervised calibration samples [5–8]. These methods utilize calibration samples with noisy or partial labels, and show that small coverage gaps can be achieved with a moderated amount of incorrect or missing labels. However, existing methods cannot handle cases with unsupervised calibration where all the calibration samples are unlabeled. In particular, it has been shown that set-prediction rules obtained using *only* unsupervised samples for calibration provide either uninformative or unreliable predictions (see Def. 2 and Thm. 3 in [6]).

This paper presents effective methods for split conformal prediction with unsupervised calibration samples. In particular, the proposed approach avoids the negative result for weakly-supervised calibration in [6] because the set-prediction rule is obtained leveraging also supervised training samples previously used to learn the classification rule. Theoretical and experimental results show that the presented methods can achieve performance comparable to that with supervised calibration (see Figure 1), at the expenses of a moderate weakening of the performance guarantees and an affordable increase in computational complexity. The main contributions presented in the paper are as follows.

- We present split conformal methods that assign label weights to unsupervised calibration samples so that they are statistically indistinguishable from training data. In particular, the methods proposed minimize a two-sample nonparametric test statistic given by a family of functions.

- We provide performance guarantees for the methods presented in terms of high-confidence bounds for coverage probabilities. In addition, the coverage gap achieved is characterized in terms of a bias-variance trade-off corresponding to the family of functions used.

- We detail the methods' implementation using families of functions given by reproducing kernel Hilbert spaces (RKHSs), which result in particularly advantageous performance guarantees that also serve to select the most adequate kernel.

- We experimentally show that the proposed methods can achieve performance comparable to that with supervised calibration.

*Notations:* Vectors and matrices are denoted by bold lower and upper case letters, respectively (e.g., $\mathbf{v}$ and $\mathbf{M}$); random variables (RVs) are distinguished from instantiations by using capital upright symbols (e.g., Z); $\preceq$ and $\succeq$ denote vector (component-wise) inequalities; $\mathbb{I}\{\cdot\}$ denotes the indicator function; for an integer $t$, $[t]$ denotes the set $[t] = \{1, 2, \ldots, t\}$; $\mathbf{1}_t$ denotes the vector of size $t$ with all components equal to 1; and $\mathbf{I}_t$ denotes the identity matrix of size $t \times t$.

## 2 Preliminaries and conventional approach for split conformal prediction

Let $\mathcal{Z} = \mathcal{X} \times \mathcal{Y}$ be the set of instance-label pairs, with $\mathcal{X} \subset \mathbb{R}^d$ the set of instances (features) and $\mathcal{Y} = \{1, 2, \ldots, c\}$ the set of labels (classes), $(X, Y)$ be an RV of instance-label pairs in $\mathcal{Z}$, and $\mathcal{D}$ be a dataset composed by i.i.d. copies of $(X, Y)$. Conformal prediction methods use the dataset $\mathcal{D}$ to find a set-prediction rule $\mathcal{C} : \mathcal{X} \to 2^{\mathcal{Y}}$, so that prediction sets $(\mathcal{C}(X) \subset \mathcal{Y})$ have reduced cardinality and often cover the true label $(Y \in \mathcal{C}(X))$. In particular, such methods aim to ensure a coverage probability near a target value $1 - \alpha$ (e.g., $\alpha = 0.1$).

The most common approach for conformal prediction is known as split conformal prediction. In this approach, the dataset is randomly split into two groups $\mathcal{D} = \mathcal{D}_{\text{train}} \cup \mathcal{D}_{\text{cal}}$: the $m$ samples $\mathcal{D}_{\text{train}} = \{(\widetilde{X}_i, \widetilde{Y}_i)\}_{i \in [m]}$ are referred to as training (or pretraining) samples, and the remaining $n$ samples $\mathcal{D}_{\text{cal}} = \{(X_i, Y_i)\}_{i \in [n]}$ are referred to as calibration samples. Then, prediction sets are obtained as shown in Algorithm 1 below (see e.g., [2]).

---

**Algorithm 1** Supervised split conformal prediction

---

**Input:** Training samples $\{(\widetilde{X}_i, \widetilde{Y}_i)\}_{i \in [m]}$, labeled calibration samples $\{(X_i, Y_i)\}_{i \in [n]}$, and coverage $1 - \alpha$
  1: Using the training samples, construct a conformal score function $S : \mathcal{X} \times \mathcal{Y} \to \mathbb{R}$
  2: Compute the conformal scores of the calibration samples $\{S(X_i, Y_i)\}_{i \in [n]}$
  3: Compute the conformal quantile $\widehat{q} = \text{Quantile}\big(\{(S(X_i, Y_i), 1/n)\}_{i \in [n]}; (1 - \alpha)(1 + 1/n)\big)$
  4: For any test instance $X$, return the prediction set $\mathcal{C}(X) = \{y \in \mathcal{Y} : S(X, y) \leq \widehat{q}\}$

---

The $m$ training samples are used to learn a classification rule that determines a conformal score $S : \mathcal{X} \times \mathcal{Y} \to \mathbb{R}$ assessing the plausibility of different instance-label pairs. For instance, the conformal score can be defined as $S(X, Y) = 1 - \widehat{p}(Y|X)$ where $\widehat{p}(Y|X)$ is the probability estimate (e.g., softmax output) assigned to label $Y$ by the classification rule at instance $X$ (see [2,9] for other types of conformal scores for classification tasks). Subsequently, the $n$ calibration samples are used to obtain score values and the corresponding conformal quantile $\widehat{q}$ in Step 3 of Algortihm 1. Specifically, the expression $\text{Quantile}\big(\{(S_i, p_i)\}_{i \in [t]}; \beta\big)$ denotes the $\beta$-quantile of values $S_1, S_2, \ldots, S_t \in \mathbb{R}$ with probabilities $p_1, p_2, \ldots, p_t \in [0, 1]$, $\sum_{i \in [t]} p_i = 1$, that is

$$\text{Quantile}\big(\{(S_i, p_i)\}_{i \in [t]}; \beta\big) = \inf \big\{q \in \mathbb{R} : \sum_{i \in [t]} p_i \mathbb{I}\{S_i \leq q\} \geq \beta\big\}. \tag{1}$$

Then, the set-prediction rule is directly obtained using the score $S$ and the conformal quantile $\widehat{q}$.

The conventional approach for split conformal prediction provides remarkably strong performance guarantees for the resulting prediction sets [1–3, 10]. Specifically, for any data distribution, type of conformal score, and target coverage $1 - \alpha$, the set-prediction rule $\mathcal{C}$ from Algorithm 1 satisfies

$$1 - \alpha \leq \ \mathbb{P}\{Y \in \mathcal{C}(X)\} \ \leq 1 - \alpha + \frac{1}{n+1} \tag{2}$$

$$1 - \alpha - \sqrt{\frac{\log(2/\delta)}{2n}} \leq \mathbb{P}\{Y \in \mathcal{C}(X)|\mathcal{D}\} \leq 1 - \alpha + \frac{1}{n+1} + \sqrt{\frac{\log(2/\delta)}{2n}}, \ \text{w.p.} \geq 1 - \delta. \tag{3}$$

These performance guarantees are satisfied with wide generality. The lower bounds for the marginal coverage and the (training) conditional coverage in (2) and (3), respectively, do not require additional assumptions. As described above, we assume that training, calibration, and test samples are randomly drawn from the same data distribution. The lower bound in (2) does not even require the independence of samples and is valid also if samples are exchangeable. The upper bounds in (2) and (3) additionally require that the score values are distinct with probability one, which can be easily achieved for instance by adding a negligible amount of noise [2]. The bounds for training-conditional coverage in (3) are arguably more relevant in a practical sense: in a given application, there is only one dataset $\mathcal{D}$, and the achieved coverage corresponds to a specific realization of $\mathbb{P}\{Y \in \mathcal{C}(X)|\mathcal{D}\}$, rather than its expected value $\mathbb{P}\{Y \in \mathcal{C}(X)\}$.

Split conformal prediction has been shown to be extremely useful in practice since it allows to obtain reliable set-prediction rules from general prediction rules (possibly highly unreliable) by using an additional set of calibration samples. However, existing methods require the use supervised calibration samples, leading to additional labeling costs. In the following, we describe the proposed approach for split conformal prediction that uses unsupervised calibration samples.

# 3 Split conformal prediction with unsupervised calibration samples

In the proposed approach, the labels of the calibration samples are not available, and prediction sets are obtained as shown in Algorithm 2 below.

---

**Algorithm 2** Split conformal prediction with unsupervised calibration

---

**Input:** Training samples $\{(\widetilde{X}_i, \widetilde{Y}_i)\}_{i \in [m]}$, unlabeled calibration samples $\{X_i\}_{i \in [n]}$, and coverage $1 - \alpha$

1: Using the training samples, construct a conformal score function $S : \mathcal{X} \times \mathcal{Y} \to \mathbb{R}$
2: Compute the conformal scores of the calibration instances for all possible labels $\{S(X_i, y)\}_{(i,y) \in [n] \times \mathcal{Y}}$
3: Obtain label weights $\{w_i(y)\}_{(i,y) \in [n] \times \mathcal{Y}}$ for the calibration instances
4: Compute the conformal quantile $\widehat{q} = \text{Quantile}\big(\{(S(X_i, y), w_i(y)/n)\}_{(i,y) \in [n] \times \mathcal{Y}}; (1 - \alpha)(1 + 1/n)\big)$
5: For any test instance $X$, return the prediction set $\mathcal{C}(X) = \{y \in \mathcal{Y} : S(X, y) \le \widehat{q}\}$

---

The main difference in the proposed approach with respect to conventional split conformal prediction consists of Step 3 in Algorithm 2 that obtains label weights for unsupervised calibration samples. Indeed, the conventional approach can be seen as a particular case of Algorithm 2 in which the label weights correspond to the actual calibration labels, i.e., $w_i(y)$ is taken as $w_i^*(y) = \mathbb{I}\{y = Y_i\}$ for any $(i, y) \in [n] \times \mathcal{Y}$.

In the presented methods, the label weights for calibration instances are obtained using the training samples. A naive approach would obtain such weights using the labels predicted by a classification rule h learned using the training samples, i.e., $w_i(y)$ taken as $\widehat{w}_i(y) = \mathbb{I}\{y = h(X_i)\}$ for any $(i, y) \in [n] \times \mathcal{Y}$. However, this naive approach only provides acceptable performance in cases where the classification rule is already highly accurate, making conformation prediction somewhat unnecessary. The following sections present methods that avoid the limitations of such naive approaches and obtain the label weights by solving a tractable optimization problem determined by a family of functions. The proposed approach can be seen as obtaining labels that result in calibration and training samples that are statistically indistinguishable by a two-sample test based on an integral probability metric (IPM). This type of non-parametric tests can reliably detect statistical differences between two sets of samples without requiring any distributional assumption (see e.g., [11, 12]).

## 3.1 Label weights minimizing IPMs

Let $\mathbf{w} = [w_1(1), w_1(2), \ldots, w_1(c), w_2(1), \ldots, w_n(c)]^\top \in \mathbb{R}^{nc}$ denote label weights for the calibration samples, the Step 3 in Algorithm 2 can be carried out by solving the optimization problem

$$\min_{\mathbf{w}} \sup_{f \in \mathcal{F}} \left| \frac{1}{n} \sum_{(i,y) \in [n] \times \mathcal{Y}} w_i(y) f(X_i, y) - \frac{1}{m} \sum_{i \in [m]} f(\widetilde{X}_i, \widetilde{Y}_i) \right| \qquad (4)$$

$$\text{s.t. } \mathbf{w} \succeq \mathbf{0}, \ \mathbf{A}\mathbf{w} = \mathbf{1}_n, \ \mathbf{B}\mathbf{w} \preceq \mathbf{b}$$

where $\mathcal{F}$ is a family of functions $f : \mathcal{X} \times \mathcal{Y} \to \mathbb{R}$. The optimization in (4) is convex for any family $\mathcal{F}$ since the supremum of convex functions is a convex function [13], and is tractable for common and general families of functions. In particular, using functions in RKHSs, optimization (4) becomes a quadratic program of size $\mathcal{O}(nc)$ (see details in Section 4 below), while using Lipschitz functions, optimization (4) becomes a linear program of size $\mathcal{O}(ncm)$, as a consequence of the Kantorovich-Rubinstein duality theorem [14].

Label weights aim to describe the probabilities of different labels, and the linear feasible set in (4) encodes quite general prior knowledge for label probabilities. In particular, the matrix $\mathbf{A} \in \mathbb{R}^{n \times nc}$ just encodes the constraints $\sum_{y \in \mathcal{Y}} w_i(y) = 1$ for $i \in [n]$, i.e., $\mathbf{A} = \mathbf{I}_n \otimes \mathbf{1}_c^\top$ for $\otimes$ the matrix Kronecker product. On the other hand, the matrix $\mathbf{B} \in \mathbb{R}^{r \times nc}$ and vector $\mathbf{b} \in \mathbb{R}^r$ can be used to encode other types of prior knowledge about label probabilities. For instance, an upper bound $L$ for the expected loss $\mathbb{E}\{\ell(h(X), Y)\}$ of a classifier h (e.g., its cross-entropy loss) can be used to have the constraint given by $\mathbf{b} = nL \in \mathbb{R}$ and

$$\mathbf{B} = [\ell(h(X_1), 1), \ell(h(X_1), 2), \ldots, \ell(h(X_1), c), \ell(h(X_2), 1), \ldots, \ell(h(X_n), c)] \in \mathbb{R}^{1 \times nc}. \qquad (5)$$

Such feasible sets often contain the label weights corresponding to the actual calibration labels, $w_i^*(y) = \mathbb{I}\{y = Y_i\}$ for $i \in [n]$, $y \in \mathcal{Y}$, because $\mathbf{w}^* \succeq \mathbf{0}$ and $\mathbf{A}\mathbf{w}^* = \mathbf{1}_n$ are clearly achieved, and we have $\mathbf{B}\mathbf{w}^* \preceq \mathbf{b}$ as long as the prior knowledge encoded by $\mathbf{B}$ and $\mathbf{b}$ is reliable.

As shown in the following, Algorithm 2 with label weights obtained by (4) can provide theoretical guarantees and practical performance comparable to those achieved by conventional split conformal prediction with supervised calibration samples. The optimization in (4) can be seen as finding the label weights for calibration samples that minimize the IPM between probability distributions corresponding to training and calibration samples, where for a family of functions $\mathcal{F}$

$$\mathrm{IPM}(P, Q) = \sup_{f \in \mathcal{F}} |\mathbb{E}_P\{f\} - \mathbb{E}_Q\{f\}|.$$

In particular, for families given by RKHSs, IPMs correspond to maximum mean discrepancy metrics while for families given by Lipschitz functions, IPMs correspond to L1-Wasserstein metrics (see e.g., [12, 15]).

The objective function in (4) denoted as $\Phi_{n,m}^{\mathcal{F}}(\mathbf{w})$ corresponds to the IPM between the distribution $\mathbf{w}/n$ supported on the $nc$ possible instance-label calibration pairs, and the empirical distribution of the $m$ training samples. IPMs between empirical distributions have been used to design non-parametric two-sample tests in which two sets of samples are considered to be statistically different (draw from different distributions) if the IPM between the corresponding empirical distributions is large enough [11, 12]. Therefore, the label weights obtained from (4) can be interpreted as those making calibration and training samples to be statistically indistinguishable by a non-parametric two-sample test. In particular, at the weights $\mathbf{w}^*$ corresponding to the actual calibration samples the value of (4) is given by the test statistic achieved by two sets of i.i.d. samples that are actually drawn from the same distribution.

A key methodological difference in the proposed approach is that the training samples are not only used to learn the score function but also to obtain label weights for calibration samples. In particular, the methods presented bypass the negative results in [6] for weakly-supervised calibration because the set prediction function is obtained using unlabeled calibration samples and also labeled training samples. This reuse of the training samples may seem doomed to fail since scores and label weights for calibration samples are no longer independent (both depend on the training samples). The following briefly describes the rationale that allows the proposed approach to avoid such concerns, Theorem 1 below provides the precise performance guarantees achieved by the methods proposed.

Let $\mathfrak{X}_{\mathcal{C}} : \mathcal{X} \times \mathcal{Y} \to \{0, 1\} \subset \mathbb{R}$ be the function

$$\mathfrak{X}_{\mathcal{C}}(x, y) \coloneqq \mathbb{I}\{y \in \mathcal{C}(x)\} = \mathbb{I}\{S(x, y) \leq \widehat{q}\} \in \{0, 1\}. \tag{6}$$

The coverage gap of the set-prediction rule from Algorithm 2 is small as long as the weighted average of function $\mathfrak{X}_{\mathcal{C}}$ at the calibration instances is near the expectation of $\mathfrak{X}_{\mathcal{C}}$ because

$$\mathbb{P}\{Y \in \mathcal{C}(X)\} = \mathbb{E}\{\mathfrak{X}_{\mathcal{C}}\} \text{ and } \frac{1}{n} \sum_{i \in [n], y \in \mathcal{Y}} w_i(y) \mathfrak{X}_{\mathcal{C}}(X_i, y) \approx 1 - \alpha$$

by definition of $\mathfrak{X}_{\mathcal{C}}$ in (6) and $\widehat{q}$ in the step 4 of Algorithm 2. Since the minimization of (4) ensures that

$$\frac{1}{n} \sum_{i \in [n], y \in \mathcal{Y}} w_i(y) f(X_i, y) \approx \frac{1}{m} \sum_{i \in [m]} f(\widetilde{X}_i, \widetilde{Y}_i), \ \forall f \in \mathcal{F}$$

prediction sets given by label weights from (4) have a small coverage gap as long as the family $\mathcal{F}$ utilized: 1) can accurately approximate the function $\mathfrak{X}_{\mathcal{C}}$ at the calibration instances (e.g., $\mathcal{F}$ can interpolate finite sets of points), and 2) ensures the uniform convergence of i.i.d. averages (e.g., $\mathcal{F}$ has reduced Rademacher complexity). This type of analysis for conformal prediction methods is novel and can be of independent interest beyond split conformal prediction with unsupervised calibration.

## 3.2 Performance guarantees

The following provides performance bounds for the coverage probability of set-prediction rules obtained using the proposed approach in Algorithm 2. In particular, such performance guarantees are given in terms of the Rademacher complexity of the family $\mathcal{F}$, and the quantity

$$D(\mathfrak{X}_{\mathcal{C}}, \mathcal{F}) = \inf_{f \in \mathcal{F}} \frac{1}{n} \sum_{i \in [n], y \in \mathcal{Y}} \left| \mathfrak{X}_{\mathcal{C}}(X_i, y) - f(X_i, y) \right| \tag{7}$$

that describes the smallest error achieved approximating the function $\mathfrak{X}_\mathcal{C}$ with functions in $\mathcal{F}$ at the calibration instances. For the next result we denote by $\mathcal{R}_t(\mathcal{F})$ the Rademacher complexity of $\mathcal{F}$ for $t$ i.i.d. samples, and consider general families of real-valued functions with bounded differences, (i.e., $|f(z) - f(z')| \leq B$, for any $f \in \mathcal{F}$ and $z, z' \in \mathcal{Z} = \mathcal{X} \times \mathcal{Y}$). The requirement of bounded differences is satisfied by common families including functions in RKHSs given by bounded kernels or Lipschitz functions defined in bounded sets.

**Theorem 1.** Let $\mathbf{w}$ be label weights with objective value in (4) upper bounded by $V_{\text{opt}}$ (i.e., $\Phi_{n,m}^\mathcal{F}(\mathbf{w}) \leq V_{\text{opt}}$). For any data distribution, type of conformal score, and target coverage $1 - \alpha$, the set-prediction rule $\mathcal{C}$ from Algorithm 2 satisfies

$$\mathbb{P}\{Y \in \mathcal{C}(X)|\mathcal{D}\} \geq 1 - \alpha - G_{n,m}^\mathcal{F} - \sqrt{\frac{\log(2/\delta)}{2n}} \qquad (8)$$

with probability at least $1 - \delta$, for $G_{n,m}^\mathcal{F}$ given by

$$G_{n,m}^\mathcal{F} = V_{\text{opt}} + D(\mathfrak{X}_\mathcal{C}, \mathcal{F}) + 2(\mathcal{R}_n(\mathcal{F}) + \mathcal{R}_m(\mathcal{F})) + B\sqrt{\left(\frac{1}{n} + \frac{1}{m}\right)\frac{\log(2/\delta)}{2}}. \qquad (9)$$

In addition, if the values $\{S(X_i, y)\}_{(i,y) \in [n] \times \mathcal{Y}}$ are distinct with probability one, we have

$$\mathbb{P}\{Y \in \mathcal{C}(X)|\mathcal{D}\} \leq 1 - \alpha + \frac{2}{n+1} + G_{n,m}^\mathcal{F} + \sqrt{\frac{\log(2/\delta)}{2n}} \qquad (10)$$

with probability at least $1 - \delta$.

*Proof.* See Appendix B. □

The result above presents performance guarantees for the coverage probability of the methods proposed comparable to those in (3) for the conventional approach with supervised calibration. The excess coverage gap $G_{n,m}^\mathcal{F}$ in the bounds of Theorem 1 describes the effectiveness of Step 3 in Algorithm 2. In particular, the term $V_{\text{opt}}$ accounts for the objective value obtained by $\mathbf{w}$ in (4), and is small using appropriate optimization algorithms and number of calibration and training samples. For instance, if the feasible set in optimization (4) includes the label weights $\mathbf{w}^*$ corresponding to the actual calibration labels, the objective value $\Phi_{n,m}^\mathcal{F}(\mathbf{w})$ satisfies

$$\Phi_{n,m}^\mathcal{F}(\mathbf{w}) \leq \varepsilon_{\text{opt}} + 2(\mathcal{R}_n(\mathcal{F}) + \mathcal{R}_m(\mathcal{F})) + B\sqrt{\left(\frac{1}{n} + \frac{1}{m}\right)\frac{\log(1/\delta)}{2}}, \quad \text{w.p.} \geq 1 - \delta \qquad (11)$$

for $\varepsilon_{\text{opt}}$ the optimization error achieved by the algorithm used to solve (4) (see Appendix D for a detailed proof). Hence, the term in (9) due to the optimization value is generally of the same order as the Rademacher complexity of the family. In addition, Theorem 1 shows that the methods proposed do not require to solve the optimization problem (4) with a high accuracy, as approximate optimization algorithms increase the bound only by their optimization error $\varepsilon_{\text{opt}}$. For instance optimization errors on the order of $10^{-3}$ do not significantly affect the final coverage probability.

The excess in coverage gap achieved using unsupervised calibration is given by the terms $D(\mathfrak{X}_\mathcal{C}, \mathcal{F})$ and $\mathcal{R}_n(\mathcal{F}) + \mathcal{R}_m(\mathcal{F})$ that describe a bias-variance trade-off for the family $\mathcal{F}$. The bias term $D(\mathfrak{X}_\mathcal{C}, \mathcal{F})$ corresponds to the distance between function $\mathfrak{X}_\mathcal{C}$ and the family $\mathcal{F}$ at the calibration instances, while the variance term $\mathcal{R}_n(\mathcal{F}) + \mathcal{R}_m(\mathcal{F})$ accounts for the usage of finite sample sizes. Both terms are low using an appropriate family $\mathcal{F}$ and a sufficient number of samples. The bias term $D(\mathfrak{X}_\mathcal{C}, \mathcal{F})$ can be made arbitrarily small using functions that are Lipschitz or belong to an RKHS given by a strictly positive kernel (e.g., Gaussian, Laplacian, or polynomial) since both types of functions can interpolate any finite set of points. The variance term $\mathcal{R}_n(\mathcal{F}) + \mathcal{R}_m(\mathcal{F})$ decreases with the sample sizes at a rate $\mathcal{O}(Rn^{-1/2} + Rm^{-1/2})$ using functions in an RKHS with norm bounded by $R$ (see e.g, Chapter 6 in [16]). On the other hand, using $R$-Lipschitz functions such a rate becomes $\mathcal{O}(Rn^{-1/d} + Rm^{-1/d})$ [17, 18] that is significantly affected by the instances' dimensionality $d$.

Theorem 1 shows that the coverage gap decreases with both the number of calibration samples $n$ and training samples $m$. Although the total number of training samples available is commonly much larger than that of calibration samples, there is not a significant benefit in using $m > n$ for calibration, and label weights can be obtained using in (4) only a subset of the training samples available. The results in the theorem also expose an important limitation in the methods presented since the performance guarantees deteriorate for an increased number of classes $|\mathcal{Y}| = c$. Specifically, both

bias and variance terms increase with the number of classes, the bias term $D(\mathfrak{X}_{\mathcal{C}}, \mathcal{F})$ is given by the average of approximation errors multiplied by $|\mathcal{Y}|$, and the variance term $\mathcal{R}_n(\mathcal{F}) + \mathcal{R}_m(\mathcal{F})$ is given by the complexity of functions over $\mathcal{X} \times \mathcal{Y}$ that generally increases with $|\mathcal{Y}|$.

The performance guarantees shown in Theorem 1 can also be compared with those of the naive approach that labels calibration instances using a classification rule. It is easy to show that for such a naive approach the coverage gap is $\mathcal{O}(\text{Err}(\text{h}))$ for $\text{Err}(\text{h})$ the error probability of the classification rule h used to label the calibration instances. In particular, such a bound for the coverage gap can be obtained by using the results in [19] since $\text{Err}(\text{h})$ coincides with the total variation distance between $(\text{X}, \text{Y})$ and $(\text{X}, \text{h}(\text{X}))$. A coverage gap near $\text{Err}(\text{h})$ is not acceptable in common scenarios because it would only be small in cases where a highly accurate classifier is available, which would commonly defeat the purpose of conformal prediction.

In this section we have described the general approach presented together with the corresponding performance guarantees. As indicated above, kernel-based methods offer an advantageous balance for optimization efficiency and bias-variance trade-off in general scenarios, so that the rest of the paper focuses in such type of approaches. In addition, the kernel-based methods provide enhanced performance guarantees that enable effective model selection, as described in the following.

## 4  Kernel-based implementation

Let $\mathcal{H}$ be an RKHS of real-valued functions over $\mathcal{Z} = \mathcal{X} \times \mathcal{Y}$ given by kernel $K : \mathcal{Z} \times \mathcal{Z} \to \mathbb{R}$. The optimization problem in (4) using family $\mathcal{F} = \{f \in \mathcal{H} : \|f\|_{\mathcal{H}} \leq R\}$ with $R = 1$ is equivalent to

$$\min_{\mathbf{w}} \left( \frac{1}{n^2} \mathbf{w}^{\top} \mathbf{K} \, \mathbf{w} - \frac{2}{nm} \mathbf{v}^{\top} \mathbf{w} + \frac{1}{m^2} \mathbf{1}_m^{\top} \widetilde{\mathbf{K}} \, \mathbf{1}_m \right)^{\frac{1}{2}} \tag{12}$$
$$\text{s.t.} \quad \mathbf{w} \succeq \mathbf{0}, \mathbf{A}\mathbf{w} = \mathbf{1}_n, \mathbf{B}\mathbf{w} \preceq \mathbf{b}$$

where $\mathbf{K} \in \mathbb{R}^{nc \times nc}$ is the kernel matrix corresponding to the $nc$ possible instance-label calibration pairs $Z_1, Z_2, \ldots, Z_{nc}$ with $Z_{(i-1)c+y} = (X_i, y)$ for $i \in [n], y \in \mathcal{Y}$, that is, the $(i, j)$-th component of $\mathbf{K}$ is given by $K_{i,j} = K(Z_i, Z_j)$ for $i, j \in [nc]$. Similarly, $\widetilde{\mathbf{K}} \in \mathbb{R}^{m \times m}$ is the kernel matrix corresponding to the training samples $\widetilde{Z}_1, \widetilde{Z}_2, \ldots, \widetilde{Z}_m$ with $\widetilde{Z}_i = (\widetilde{X}_i, \widetilde{Y}_i)$ for $i \in [m]$. In addition, $\mathbf{v} \in \mathbb{R}^{nc}$ is the vector with $i$-th component given by $v_i = \sum_{j \in [m]} K(Z_i, \widetilde{Z}_j)$ for $i \in [nc]$. The equivalence between (4) and (12) follows as a consequence of the dual characterization of the norm in $\mathcal{H}$ (see the detailed derivation in Appendix D). The kernel $K$ can be any joint kernel over $\mathcal{Z} = \mathcal{X} \times \mathcal{Y}$, for instance $K$ can be given by a kernel $K_0$ over $\mathcal{X}$ as the separable kernel $K\big((x, y), (x', y')\big) = K_0(x, x')\mathbb{I}\{y = y'\}$ (see [20] for other types of joint kernels). Separable kernels are simple and convenient computationally since they lead to highly sparse kernel matrices.

The usage of a family with other maximum norm $R \neq 1$ results in an optimization problem equivalent to (12), namely, a problem that has the same solution with optimum value multiplied by $R$. This fact has important consequences because for families of functions $\mathcal{F} = \{f \in \mathcal{H} : \|f\|_{\mathcal{H}} \leq R\}$ the bias/variance trade-off described in Section 3.2 is given by the value of the norm $R$. Increased values of $R$ result in more general families of functions at the expenses of an increased Rademacher complexity. Therefore, kernel-based methods provide an automatic capacity/complexity control: the trade-off for the value of $R$ does not need to be explicitly addressed because the optimization problem solved does not really change with $R$. Therefore, a solution of (12) enjoys the performance guarantees of Theorem 1 corresponding to the value of $R$ that achieves the best bias/variance trade-off. In particular, kernel-based methods result in the following enhanced performance guarantees.

**Theorem 2.** Let $\mathcal{H}$ be an RKHS given by kernel $K$ over $\mathcal{Z} = \mathcal{X} \times \mathcal{Y}$ with $0 \leq K(z, z') \leq \kappa^2$ $\forall z, z' \in \mathcal{Z}$, and $\mathbf{w}$ be a solution of optimization problem (12) with feasible set that includes the label weights $\mathbf{w}^*$ corresponding to the actual calibration labels. For any data distribution, type of conformal score, and target coverage $1 - \alpha$, the set-prediction rule $\mathcal{C}$ from Algorithm 2 satisfies

$$\mathbb{P}\{Y \in \mathcal{C}(X) | \mathcal{D}\} \geq 1 - \alpha - G_{n,m}^K - \sqrt{\frac{\log(2/\delta)}{2n}} \tag{13}$$

with probability at least $1 - \delta$, for $G_{n,m}^K$ given by

$$G_{n,m}^K = \min_{f \in \mathcal{H}} \frac{1}{n} \sum_{i \in [n], y \in \mathcal{Y}} \left| \mathfrak{X}_{\mathcal{C}}(X_i, y) - f(X_i, y) \right| + 2\kappa\big(1 + \sqrt{\log(2/\delta)}\big)\sqrt{\frac{1}{n} + \frac{1}{m}} \, \|f\|_{\mathcal{H}}. \tag{14}$$

In addition, if the values $\{S(X_i, y)\}_{(i,y)\in[n]\times\mathcal{Y}}$ are distinct with probability one, we have

$$\mathbb{P}\{Y \in \mathcal{C}(X)|\mathcal{D}\} \leq 1 - \alpha + \frac{2}{n+1} + G_{n,m}^K + \sqrt{\frac{\log(2/\delta)}{2n}} \tag{15}$$

with probability at least $1 - \delta$.

*Proof.* See Appendix C. □

The result above shows that the kernel-based approach in (12) yields an excess coverage gap given by the best bias-variance trade-off in the corresponding RKHS. This best trade-off is described by the term $G_{n,m}^K$ in (14) as a combination of the approximation error and the RKHS norm weighted by the number of samples used. For simplicity, Theorem 2 is stated for the case where $\mathbf{w}$ is an exact solution of (12) and $\mathbf{w}^*$ is included in the feasible set. A similar result holds in general cases, by including the term $V_{\text{opt}}$ in the expression of $G_{n,m}^K$ to account for the optimization value attained, as in Theorem 1.

For RKHSs given by strictly positive definite kernels, Theorem 2 shows that the coverage gap is $\mathcal{O}(M\sqrt{1/n + 1/m})$ where $M$ denotes the minimum norm of interpolators of $\mathfrak{X}_\mathcal{C}$ at the calibration instances. More generally, the excess coverage gap can be bounded as $G_{n,m}^K \leq c\varepsilon + 2\kappa(1 + \sqrt{\log(2/\delta)})M_\varepsilon\sqrt{1/n + 1/m}$ where $M_\varepsilon$ denotes the minimum norm of $\varepsilon$-approximations of $\mathfrak{X}_\mathcal{C}$ at the calibration instances, that is

$$M_\varepsilon = \min_{f\in\mathcal{H}}\{\|f\|_\mathcal{H} : |\mathfrak{X}_\mathcal{C}(X_i, y) - f(X_i, y)| \leq \varepsilon, \ i \in [n], y \in \mathcal{Y}\}. \tag{16}$$

The magnitude of the minimum norm $M$ (or $M_\varepsilon$) depends on the smoothness of the function $\mathfrak{X}_\mathcal{C}$ at the calibration instances and on how well the selected kernel is adapted to such values. The smoothness of $\mathfrak{X}_\mathcal{C}$ is determined by that of the conformal score function $S(X, Y)$ defined by the classification rule used. Such a function may be highly non-smooth at the training instances used to learn the classification rule. However, for classifiers that generalize well, the conformal score is expected to be fairly smooth at the calibration instances, which are drawn independently of the training samples. Cases in which the conformal score is highly non-smooth across the instance space are possible but of reduced practical relevance, as such conformal scores would be largely uninformative.

The bounds in Theorem 2 not only provide theoretical guarantees but can also be used in practice for kernel selection. In particular, a kernel can be chosen from a candidate set (e.g., multiple choices of bandwidth for a Gaussian kernel) as the one achieving the smallest minimum norm interpolation (or $\varepsilon$-approximation). The minimum norm of interpolating functions can be efficiently computed using standard properties of RKHSs (see e.g., [21]). Specifically, for kernel $K$ the minimum norm of interpolators for $\mathfrak{X}_\mathcal{C}$ at the calibration instances is given by $\mathbf{u}^\top\boldsymbol{\gamma}$ for $\boldsymbol{\gamma}$ that solves the linear system $\mathbf{K}\boldsymbol{\gamma} = \mathbf{u}$, where the kernel matrix $\mathbf{K}$ and vector $\mathbf{u}$ are given by $K_{i,j} = K(Z_i, Z_j)$ and $u_i = \mathfrak{X}_\mathcal{C}(Z_i)$, for $i, j \in [nc]$. Notice also that the preprocessing step that selects the most suitable kernel over $s$ candidates only change the term $\log(2/\delta)$ in the bounds of Theorem 2 to the term $\log(2s/\delta)$, as a direct consequence of the union bound.

Algorithm 3 details the implementation of the proposed kernel-based methods to obtain label weights for the calibration instances. The kernel selection process requires to solve $s$ linear systems of equations of size $nc$ so that it has time complexity $\mathcal{O}(sn^3c^3)$. Then, label weights are obtained by solving a quadratic program with $nc$ variables and $n + r$ linear constraints. Highly accurate solutions of such problems can be obtained using interior point methods with complexity per iteration $\mathcal{O}(n^3c^3)$ due to solving a linear system (see Chapter 16 in [22]). These methods allow to efficiently solve the quadratic program for a number classes $c$ and unlabeled calibration instances $n$ in the order of tens and thousands, respectively. For cases with a higher number of classes or calibration samples, first-order methods can result in efficient implementations since their complexity per iteration is $\mathcal{O}(n^2c^2)$ due to a matrix-vector multiplication (see Chapter 16 in [22]). The memory complexity of Algorithm 3 is $\mathcal{O}(n^2c^2)$ corresponding to the storage of a kernel matrix. Furthermore, the computational complexity of Algorithm 3 can be significantly reduced taking into account the common sparsity of the matrices involved. The matrix $\mathbf{A}$ corresponding to the $n$ linear equality constraints is always highly sparse (ratio of nonzero values $1/n$), and the kernel matrix $\mathbf{K}$ is also highly sparse using separable kernels (ratio of nonzero values $1/c^2$). Appendix E shows that Algorithm 3

using interior point methods achieves running times ranging from tens to hundreds of seconds for cases with up to 10 classes and up to 3,000 calibration samples.

---

**Algorithm 3** Kernel-based method to obtain label weights for calibration instances

---

**Input:** Unsupervised calibration samples $\{X_i\}_{i\in[n]}$, training samples $\{(\widetilde{X}_i, \widetilde{Y}_i)\}_{i\in[m]}$

Kernels $K^{(1)}, K^{(2)}, \ldots, K^{(s)}$, matrix and vector for inequality constraints $\mathbf{B}$ and $\mathbf{b}$

**Output:** Label weights $w_i(y), i \in [n], y \in \mathcal{Y}$

1: $Z_{(i-1)c+y} \leftarrow (X_i, y)$ for $i \in [n], y \in \mathcal{Y}, \widetilde{Z}_i \leftarrow (\widetilde{X}_i, \widetilde{Y}_i)$ for $i \in [m]$

————————————————————— *Kernel selection* —————————————————————

2: Compute a rough approximate $\widehat{q}_0$ of the quantile using Algorithm 2 with the naive approach given by label weights $\{\widehat{w}_i(y) = \mathbb{I}\{y = \mathrm{h}(X_i)\}\}$

3: Compute vector $\mathbf{u} \in \mathbb{R}^{nc}$ given by $u_i = \mathfrak{X}_0(Z_i) = \mathbb{I}\{S(Z_i) \leq \widehat{q}_0\}$ for $i \in [nc]$

4: **for** $t \in [s]$ **do**

5:     Compute kernel matrix $\mathbf{K}^{(t)} \in \mathbb{R}^{nc\times nc}$ given by $K_{i,j}^{(t)} = K^{(t)}(Z_i, Z_j)$ for $i, j \in [nc]$

6:     $\boldsymbol{\gamma}^{(t)} \leftarrow$ solution of $\mathbf{K}^{(t)}\boldsymbol{\gamma}^{(t)} = \mathbf{u}$

7: **end for**

8: $t^* \leftarrow \underset{t\in[s]}{\arg\min}\{\mathbf{u}^\top\boldsymbol{\gamma}^{(t)}\}$

————————————————————— *Label weights computation* —————————————————————

9: $\mathbf{K} \leftarrow \mathbf{K}^{(t^*)}, \mathbf{A} \leftarrow \mathbf{I}_n \otimes \mathbf{1}_c^\top$

10: Compute vector $\mathbf{v} \in \mathbb{R}^{nc}$ given by $v_i = \sum_{j\in[m]} K^{(t^*)}(Z_i, \widetilde{Z}_j)$ for $i \in [nc]$

11: $\mathbf{w} \leftarrow$ solution of

$$\min_{\mathbf{w}} \frac{1}{n}\mathbf{w}^\top\mathbf{K}\mathbf{w} - \frac{2}{m}\mathbf{v}^\top\mathbf{w}$$

$$\text{s.t.} \quad \mathbf{w} \succeq \mathbf{0}, \mathbf{A}\mathbf{w} = \mathbf{1}_n, \mathbf{B}\mathbf{w} \preceq \mathbf{b}$$

---

## 5 Numerical results

The experiments assess the performance obtained by the proposed approach for split conformal prediction with unsupervised classification using 9 common benchmark datasets. The coverage probabilities and prediction set sizes are compared with those provided by the conventional approach that uses supervised calibration samples and by the naive approach that uses label predictions for calibration samples. Due to the amount of theoretical results presented, this section is necessarily concise. The code implementing the methods presented and reproducing the experiments can be found at `https://github.com/MachineLearningBCAM/Unsupervised-conformal-prediction-NeurIPS2025`. The supplementary materials provide implementation details and additional results in Appendix E, including running time assessments, as well as results for different target coverages and types of conformal scores.

For each dataset, 400 random realizations are generated by randomly partitioning the samples in training, calibration, and test sets. The training samples are used to learn classification rules: random forests for tabular datasets and neural networks for image datasets (fine-tuned Resnet50 for 'CIFAR10' and 'ImageNet10' datasets). The results in the main paper use the adaptive conformal score proposed in [2, 9], and the Appendix E reports results with the conformal score directly given by probability estimates [1]. Set-prediction rules are obtained from the scoring functions using $n$ calibration samples for multiple values of $n$, and coverage probabilities and prediction sets sizes are then computed using the test samples. The presented methods are implemented using Algorithm 2 with label weights obtained by Algorithm 3 using $m = n$ training samples randomly drawn from the samples used to learn the classification rule, and a Gaussian kernel selected from a set given by 10 candidate bandwidth (scale) parameters. Full implementation details are provided in Appendix E.

Table 1 shows the coverage probability and prediction set size obtained in the 9 datasets using $n = 1,000$ calibration samples for a target coverage of $1 - \alpha = 0.9$. The results in the table indicate that the performance of the methods presented is only slightly inferior to that of conventional methods with supervised calibration. In accordance with the discussion in Section 3.2 regarding the weakened guarantees in scenarios with many classes, 'Letter' –which includes 26 classes– is the only dataset showing significantly poorer performance. The table also shows that the naive approach

Table 1: Coverage probability and prediction set size for conventional approach with supervised calibration ('Supervised'), proposed approach with unsupervised calibration ('Unsupervised'), and naive unsupervised approach ('Unsup. Naive'). Results are shown as: mean (interquartile interval), and correspond to $1 - \alpha = 0.9$.

| Dataset | Coverage probability | | | Prediction set size | |
|---|---|---|---|---|---|
| | Supervised | Unsupervised | Unsup. Naive | Supervised | Unsupervised |
| Drybean | 0.90 (0.89,0.91) | 0.91 (0.90,0.91) | 0.84 (0.83,0.85) | 1.23 (1.20,1.25) | 1.25 (1.22,1.27) |
| Forestcov | 0.90 (0.89,0.91) | 0.89 (0.89,0.90) | 0.62 (0.61,0.63) | 1.90 (1.87,1.94) | 1.87 (1.85,1.89) |
| Satellite | 0.90 (0.89,0.91) | 0.90 (0.89,0.91) | 0.84 (0.83,0.85) | 1.45 (1.42,1.48) | 1.44 (1.41,1.48) |
| USPS | 0.90 (0.89,0.91) | 0.90 (0.89,0.91) | 0.88 (0.87,0.89) | 1.11 (1.09,1.13) | 1.11 (1.09,1.14) |
| MNIST | 0.90 (0.89,0.91) | 0.90 (0.90,0.91) | 0.86 (0.85,0.87) | 1.19 (1.16,1.21) | 1.20 (1.18,1.23) |
| Fashion | 0.90 (0.89,0.91) | 0.91 (0.90,0.91) | 0.80 (0.79,0.81) | 1.43 (1.39,1.47) | 1.45 (1.42,1.48) |
| CIFAR10 | 0.90 (0.89,0.91) | 0.91 (0.90,0.92) | 0.88 (0.87,0.89) | 1.00 (0.98,1.01) | 1.01 (0.99,1.02) |
| Imagnet10 | 0.90 (0.89,0.91) | 0.90 (0.89,0.91) | 0.86 (0.85,0.87) | 1.65 (1.66,1.70) | 1.62 (1.57,1.68) |
| Letter | 0.90 (0.89,0.91) | 0.85 (0.84,0.87) | 0.69 (0.68,0.71) | 5.58 (5.24,5.90) | 4.14 (3.80,4.48) |

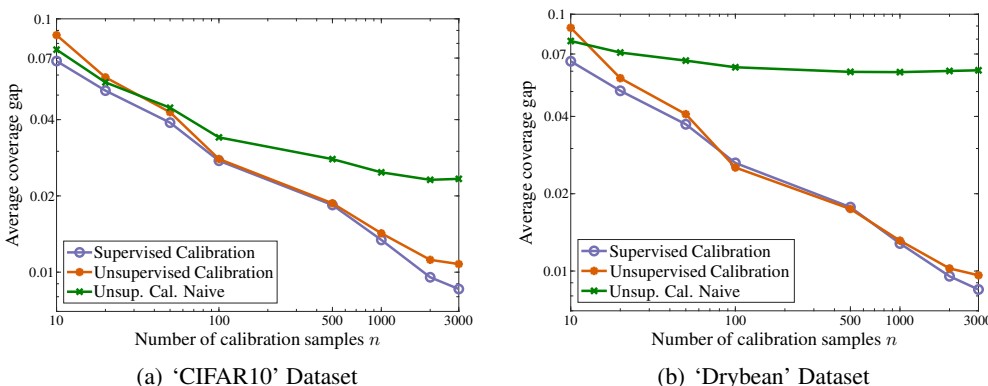

(a) 'CIFAR10' Dataset      (b) 'Drybean' Dataset

Figure 2: Decrease of average coverage gap with the number of calibration samples.

can only provide acceptable performance in cases where the learned classification rule is highly accurate (e.g., the classification error in 'CIFAR10' is $4\%$). Figure 2 shows the average coverage gap ($\mathbb{E}\{|1 - \alpha - \mathbb{P}\{Y \in \mathcal{C}(X)|\mathcal{D}\}|\}$ with $\alpha = 0.1$) obtained with different number of calibration samples. The results in the figure exhibit that coverage gaps smaller than $0.015$ can be obtained with few thousand unsupervised calibration samples. In accordance with the theoretical results in Section 3.2, the coverage gap decreases with the number of unsupervised calibration samples comparably to cases with supervised calibration.

## 6   Conclusion

The paper presents methods for split conformal classification with unsupervised calibration. In the proposed approach, label weights for calibration samples are obtained so that training and calibration samples become statistically indistinguishable by a non-parametric two-sample test. The theoretical and experimental results show that the methods presented can achieve performance comparable to that with supervised calibration, at the expenses of a moderate degradation in performance guarantees and computational efficiency. The methodology presented can lead to methods that remove the need to acquire labeled examples at the calibration stage, thus making conformal prediction more cost-efficient in general and viable even when new labels cannot be obtained after training.

**Limitations:** The proposed methodology is broadly effective but does not scale well with the number of classes, as discussed in the theoretical and experimental results. As shown in Sections 3 and 4, the excess coverage gap in the methods presented decreases at a rate $\mathcal{O}(1/\sqrt{n} + 1/\sqrt{m})$ in general scenarios, but the constants can be large if the conformal score used is highly non-smooth at the calibration samples. Such cases are rare since the score function is learned without calibration samples, and globally non-smooth scores are largely uninformative in the first place. The other main limitation of the methods is a significantly higher computational cost compared to conventional split conformal methods, which only require computing one quantile. This drawback is of limited relevance in practice since the computational cost remains small relative to typical learning processes.

## Acknowledgements

Funding in direct support of this work has been provided by project PID2022-137063NB-I00 funded by MCIN/AEI/10.13039/501100011033 and the European Union "NextGenerationEU"/PRTR, BCAM Severo Ochoa accreditation CEX2021-001142-S/MICIN/AEI/10.13039/501100011033 funded by the Ministry of Science and Innovation (Spain), and program BERC-2022-2025 funded by the Basque Government.

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

# Appendices

## A   Auxiliary lemmas

**Lemma 1.** For $i \in [n]$, let $(X_i, Y_i)$ be an instance-label pair in $\mathcal{X} \times \mathcal{Y}$, and $w_i(1), w_i(2), \ldots, w_i(c)$ be positive label weights that add to one. If $\widehat{q}$ is the quantile

$$\widehat{q} = \text{Quantile}\big(\{(S(X_i, y), w_i(y)/n))\}_{(i,y)\in[n]\times\mathcal{Y}}; (1-\alpha)(1+1/n)\big) \tag{17}$$

for some $\alpha \in [0, 1]$, and $E$ is given by

$$E = \frac{1}{n} \sum_{i \in [n]} \mathfrak{X}_{\mathcal{C}}(X_i, Y_i) - \frac{1}{n} \sum_{i\in[n],y\in\mathcal{Y}} w_i(y)\mathfrak{X}_{\mathcal{C}}(X_i, y)$$

for the function $\mathfrak{X}_{\mathcal{C}}$ defined in (6). Then, we have

$$\widehat{q} \geq \text{Quantile}\big(\{(S(X_i, Y_i), 1/n)\}_{i\in[n]}; (1-\alpha + nE/(n+1))(1+1/n)\big). \tag{18}$$

In addition, if the values $\{S(X_i, y)\}_{(i,y)\in[n]\times\mathcal{Y}}$ are distinct, we have

$$\widehat{q} \leq \text{Quantile}\big(\{(S(X_i, Y_i), 1/n)\}_{i\in[n]}; (1-\alpha + 1/(n+1) + nE/(n+1))(1+1/n)\big). \tag{19}$$

*Proof.* From the definition of $\widehat{q}$ in (17), we have

$$\frac{1}{n} \sum_{i\in[n],y\in\mathcal{Y}} w_i(y)\mathfrak{X}_{\mathcal{C}}(X_i, y) = \frac{1}{n} \sum_{i\in[n],y\in\mathcal{Y}} w_i(y)\mathbb{I}\{S(X_i, y) \leq \widehat{q}\} \geq \frac{(n+1)(1-\alpha)}{n} \tag{20}$$

then, we directly get

$$\frac{1}{n} \sum_{i=1}^{n} \mathfrak{X}_{\mathcal{C}}(X_i, Y_i) = \frac{1}{n} \sum_{i=1}^{n} \mathbb{I}\{S(X_i, Y_i) \leq \widehat{q}\}$$

$$= \frac{1}{n} \sum_{i\in[n],y\in\mathcal{Y}} w_i(y)\mathbb{I}\{S(X_i, y) \leq \widehat{q}\} + E \geq \frac{(n+1)(1-\alpha + \frac{nE}{n+1})}{n} \tag{21}$$

that gives (18).

If the values $\{S(X_i, y)\}_{(i,y)\in[n]\times\mathcal{Y}}$ are distinct, by the definition of $\widehat{q}$ in (17) we have

$$\frac{1}{n} \sum_{i\in[n],y\in\mathcal{Y}} w_i(y)\mathfrak{X}_{\mathcal{C}}(X_i, y) = \frac{1}{n} \sum_{i\in[n],y\in\mathcal{Y}} w_i(y)\mathbb{I}\{S(X_i, y) \geq \widehat{q}\} \leq \frac{(n+1)(1-\alpha)}{n} + \frac{1}{n} \tag{22}$$

because $0 \leq w_i(y) \leq 1$ for any $y \in \mathcal{Y}, i \in [n]$. Then, analogously to the previous case we get

$$\frac{1}{n} \sum_{i=1}^{n} \mathfrak{X}_{\mathcal{C}}(X_i, Y_i) = \frac{1}{n} \sum_{i=1}^{n} \mathbb{I}\{S(X_i, Y_i) \leq \widehat{q}\} \tag{23}$$

$$\leq \frac{(n+1)(1-\alpha + \frac{nE}{n+1})}{n} + \frac{1}{n} = \frac{(n+1)(1-\alpha + \frac{1}{n+1} + \frac{nE}{n+1})}{n} \tag{24}$$

that together with (21) gives (19). $\qquad\square$

**Lemma 2.** Let $\mathbf{w} = [w_1(1), w_1(2), \ldots, w_1(c), w_2(1), \ldots, w_n(c)]^\top \in \mathbb{R}^{nc}$ be label weights for the calibration samples that satisfy $\Phi_{n,m}^{\mathcal{F}}(\mathbf{w}) \leq V_{\text{opt}}$, and $\mathcal{F}$ be a family of real-valued functions with bounded differences, i.e., $|f(z) - f(z')| \leq B$, for any $f \in \mathcal{F}$ and $z, z' \in \mathcal{Z} = \mathcal{X} \times \mathcal{Y}$.

If E is the RV

$$\text{E} = \frac{1}{n} \sum_{i \in [n]} \mathfrak{X}_{\mathcal{C}}(\text{X}_i, \text{Y}_i) - \frac{1}{n} \sum_{i\in[n],y\in\mathcal{Y}} w_i(y)\mathfrak{X}_{\mathcal{C}}(\text{X}_i, y).$$

Then, with probability at least $1 - \delta$ we have

$$|\mathrm{E}| \leq V_{\mathrm{opt}} + D(\mathfrak{X}_{\mathcal{C}}, \mathcal{F}) + 2(\mathcal{R}_n(\mathcal{F}) + \mathcal{R}_m(\mathcal{F})) + B\sqrt{\left(\frac{1}{n} + \frac{1}{m}\right)\frac{\log(1/\delta)}{2}}. \tag{25}$$

Let $\mathcal{H}$ be the RKHS given by kernel $K$ that satisfies $0 \leq K(z, z') \leq \kappa^2$ for any $z, z' \in \mathcal{Z}$. If $\mathbf{w}$ is a solution of (4) using the family $\mathcal{F}_R = \{f \in \mathcal{H} : \|f\|_{\mathcal{H}} \leq R\}$, and the weights $\mathbf{w}^*$ corresponding to the actual calibration labels are included in the feasible set of (4), we have

$$|\mathrm{E}| \leq D(\mathfrak{X}_{\mathcal{C}}, \mathcal{F}_R) + 2\kappa\left(1 + \sqrt{\log(1/\delta)}\right)R\sqrt{\frac{1}{n} + \frac{1}{m}}, \quad \text{for all } R > 0 \tag{26}$$

with probability at least $1 - \delta$.

*Proof.* For any $f \in \mathcal{F}$ we have

$$
\begin{aligned}
|\mathrm{E}| &= \left|\frac{1}{n}\sum_{i \in [n]}\mathfrak{X}_{\mathcal{C}}(\mathrm{X}_i, \mathrm{Y}_i) - \frac{1}{n}\sum_{i \in [n], y \in \mathcal{Y}} w_i(y)\mathfrak{X}_{\mathcal{C}}(\mathrm{X}_i, y)\right| \\
&= \left|\frac{1}{n}\sum_{i \in [n], y \in \mathcal{Y}}\left(\mathbb{I}\{y = \mathrm{Y}_i\} - w_i(y)\right)\mathfrak{X}_{\mathcal{C}}(\mathrm{X}_i, y)\}\right| \\
&\leq \left|\frac{1}{n}\sum_{i \in [n], y \in \mathcal{Y}}\left(\mathbb{I}\{y = \mathrm{Y}_i\} - w_i(y)\right)\left(\mathfrak{X}_{\mathcal{C}}(\mathrm{X}_i, y)\} - f(\mathrm{X}_i, y)\right)\right| \tag{27} \\
&\quad + \left|\frac{1}{n}\sum_{i \in [n], y \in \mathcal{Y}}\left(\mathbb{I}\{y = \mathrm{Y}_i\} - w_i(y)\right)f(\mathrm{X}_i, y)\right|. \tag{28}
\end{aligned}
$$

We bound separately the two terms in (27) and (28). For (27) we directly have

$$\left|\frac{1}{n}\sum_{i \in [n], y \in \mathcal{Y}}\left(\mathbb{I}\{y = \mathrm{Y}_i\} - w_i(y))\right)\left(\mathfrak{X}_{\mathcal{C}}(\mathrm{X}_i, y) - f(\mathrm{X}_i, y)\right)\right| \leq \frac{1}{n}\sum_{i \in [n], y \in \mathcal{Y}}\left|\mathfrak{X}_{\mathcal{C}}(\mathrm{X}_i, y) - f(\mathrm{X}_i, y)\right| \tag{29}$$

because $0 \leq w_i(y) \leq 1$ for any $i \in [n]$ and $y \in \mathcal{Y}$. For (28), we have

$$
\begin{aligned}
&\left|\frac{1}{n}\sum_{i \in [n], y \in \mathcal{Y}}\left(\mathbb{I}\{y = \mathrm{Y}_i\} - w_i(y)\right)f(\mathrm{X}_i, y)\right| \\
&= \left|\frac{1}{n}\sum_{i \in [n]}f(\mathrm{X}_i, \mathrm{Y}_i) - \frac{1}{m}\sum_{i \in [m]}f(\widetilde{\mathrm{X}}_i, \widetilde{\mathrm{Y}}_i) + \frac{1}{m}\sum_{i \in [m]}f(\widetilde{\mathrm{X}}_i, \widetilde{\mathrm{Y}}_i) - \frac{1}{n}\sum_{i \in [n], y \in \mathcal{Y}}w_i(y)f(\mathrm{X}_i, y)\right| \\
&\leq \left|\frac{1}{n}\sum_{i \in [n]}f(\mathrm{X}_i, \mathrm{Y}_i) - \frac{1}{m}\sum_{i \in [m]}f(\widetilde{\mathrm{X}}_i, \widetilde{\mathrm{Y}}_i)\right| + \left|\frac{1}{m}\sum_{i \in [m]}f(\widetilde{\mathrm{X}}_i, \widetilde{\mathrm{Y}}_i) - \frac{1}{n}\sum_{i \in [n], y \in \mathcal{Y}}w_i(y)f(\mathrm{X}_i, y)\right| \\
&\leq \sup_{f \in \mathcal{F}}\left|\frac{1}{n}\sum_{i \in [n]}f(\mathrm{X}_i, \mathrm{Y}_i) - \frac{1}{m}\sum_{i \in [m]}f(\widetilde{\mathrm{X}}_i, \widetilde{\mathrm{Y}}_i)\right| + \Phi_{n,m}^{\mathcal{F}}(\mathbf{w}) \tag{30}
\end{aligned}
$$

Therefore, we have

$$|\mathrm{E}| \leq \Phi_{n,m}^{\mathcal{F}}(\mathbf{w}) + D(\mathfrak{X}_{\mathcal{C}}, \mathcal{F}) + \sup_{f \in \mathcal{F}}\left|\frac{1}{n}\sum_{i \in [n]}f(\mathrm{X}_i, \mathrm{Y}_i) - \frac{1}{m}\sum_{i \in [m]}f(\widetilde{\mathrm{X}}_i, \widetilde{\mathrm{Y}}_i)\right|. \tag{31}$$

Then, the bound in (25) follows using that $\Phi_{n,m}^{\mathcal{F}}(\mathbf{w}) \leq V_{\mathrm{opt}}$, and upper bounding the 'sup' in (31). Specifically, to bound the expectation of the 'sup' in (31) we use a usual argument based on symmetrization. If $(\bar{\mathrm{X}}_1, \bar{\mathrm{Y}}_1), (\bar{\mathrm{X}}_2, \bar{\mathrm{Y}}_2), \ldots, (\bar{\mathrm{X}}_t, \bar{\mathrm{Y}}_t)$ for $t = \max(n, m)$ form a different set of i.i.d.

samples, we have

$$\mathbb{E}\Big\{\sup_{f\in\mathcal{F}}\Big|\frac{1}{n}\sum_{i\in[n]}f(\mathrm{X}_i,\mathrm{Y}_i)-\frac{1}{m}\sum_{i\in[m]}f(\widetilde{\mathrm{X}}_i,\widetilde{\mathrm{Y}}_i)\Big|\Big\}$$

$$=\mathbb{E}\Big\{\sup_{f\in\mathcal{F}}\Big|\frac{1}{n}\sum_{i\in[n]}f(\mathrm{X}_i,\mathrm{Y}_i)-\mathbb{E}\Big\{\frac{1}{n}\sum_{i\in[n]}f(\bar{\mathrm{X}}_i,\bar{\mathrm{Y}}_i)\Big\}$$

$$+\mathbb{E}\Big\{\frac{1}{m}\sum_{i\in[m]}f(\bar{\mathrm{X}}_i,\bar{\mathrm{Y}}_i)\Big\}-\frac{1}{m}\sum_{i\in[m]}f(\widetilde{\mathrm{X}}_i,\widetilde{\mathrm{Y}}_i)\Big|\Big\}$$

$$\leq\mathbb{E}\Big\{\sup_{f\in\mathcal{F}}\Big|\frac{1}{n}\sum_{i\in[n]}f(\mathrm{X}_i,\mathrm{Y}_i)-f(\bar{\mathrm{X}}_i,\bar{\mathrm{Y}}_i)\Big|\Big\}+\mathbb{E}\Big\{\sup_{f\in\mathcal{F}}\Big|\frac{1}{m}\sum_{i\in[m]}f(\bar{\mathrm{X}}_i,\bar{\mathrm{Y}}_i)-f(\mathrm{X}_i,\mathrm{Y}_i)\Big|\Big\}\quad(32)$$

$$=\mathbb{E}\Big\{\sup_{f\in\mathcal{F}}\Big|\frac{1}{n}\sum_{i\in[n]}\sigma_i(f(\mathrm{X}_i,\mathrm{Y}_i)-f(\bar{\mathrm{X}}_i,\bar{\mathrm{Y}}_i))\Big|\Big\}$$

$$+\mathbb{E}\Big\{\sup_{f\in\mathcal{F}}\Big|\frac{1}{m}\sum_{i\in[m]}\sigma_i(f(\bar{\mathrm{X}}_i,\bar{\mathrm{Y}}_i)-f(\mathrm{X}_i,\mathrm{Y}_i))\Big|\Big\}$$

$$\leq 2(\mathcal{R}_n(\mathcal{F})+\mathcal{R}_m(\mathcal{F}))\qquad(33)$$

where in (32) we have used Jensen's and triangular inequalities, and $\sigma_1,\sigma_2,\ldots,\sigma_k$ are i.i.d. Rademacher variables $\sigma_i\in\{-1,1\}$. Then, using McDiarmid's inequality (see e.g., [16]), with probability at least $1-\delta$ we get

$$\sup_{f\in\mathcal{F}}\Big|\frac{1}{n}\sum_{i\in[n]}f(\mathrm{X}_i,\mathrm{Y}_i)-\frac{1}{m}\sum_{i\in[m]}f(\widetilde{\mathrm{X}}_i,\widetilde{\mathrm{Y}}_i)\Big|\leq 2(\mathcal{R}_n(\mathcal{F})+\mathcal{R}_m(\mathcal{F}))+B\sqrt{\frac{1}{2}\Big(\frac{1}{n}+\frac{1}{m}\Big)\log\frac{1}{\delta}}$$

$$(34)$$

because changing any $(\mathrm{X}_i,\mathrm{Y}_i)$ or $(\widetilde{\mathrm{X}}_i,\widetilde{\mathrm{Y}}_i)$ by other sample in the left hand side of (34), changes the 'sup' at most $B/n$ or $B/m$, since $|f(x,y)-f(x',y')|\leq B$, for any $f\in\mathcal{F}$ and pairs $(x,y),(x',y')$. Therefore, the bound in (25) is directly obtained from (31) and (34).

The bound in (26) for families in RKHSs is obtained similarly from (30). The term corresponding to maximum deviations of empirical averages can be bounded more tightly in the RKHS case, without resorting to Rademacher complexities, as follows. Firstly, for the case $R=1$ we have

$$\sup_{f\in\mathcal{F}_1}\Big|\frac{1}{n}\sum_{i\in[n]}f(\mathrm{X}_i,\mathrm{Y}_i)-\frac{1}{m}\sum_{i\in[m]}f(\widetilde{\mathrm{X}}_i,\widetilde{\mathrm{Y}}_i)\Big|$$

$$=\sup_{f\in\mathcal{H},\|f\|_{\mathcal{H}}\leq 1}\Big|\frac{1}{n}\sum_{i\in[n]}f(\mathrm{X}_i,\mathrm{Y}_i)-\frac{1}{m}\sum_{i\in[m]}f(\widetilde{\mathrm{X}}_i,\widetilde{\mathrm{Y}}_i)\Big|$$

$$=\Big\|\frac{1}{n}\sum_{i\in[n]}K(\mathrm{X}_i,\mathrm{Y}_i,\cdot)-\frac{1}{m}\sum_{i\in[m]}K(\widetilde{\mathrm{X}}_i,\widetilde{\mathrm{Y}}_i,\cdot)\Big\|_{\mathcal{H}}\qquad(35)$$

as a consequence of the dual characterization of the norm in $\mathcal{H}$. In addition, the expression in (35) also bounds $\Phi_{n,m}^{\mathcal{F}_1}(\mathbf{w})$ since $\mathbf{w}$ is solution of (12) and $\mathbf{w}^*$ is feasible in (12).

The expectation of (35) can be bounded using Jensen's inequality $(\mathbb{E}\{s\}\leq\sqrt{\mathbb{E}\{s^2\}})$ and the bound

$$\mathbb{E}\Big\{\Big\|\frac{1}{n}\sum_{i\in[n]}K(\mathrm{X}_i,\mathrm{Y}_i,\cdot)-\frac{1}{m}\sum_{i\in[m]}K(\widetilde{\mathrm{X}}_i,\widetilde{\mathrm{Y}}_i,\cdot)\Big\|_{\mathcal{H}}^2\Big\}$$

$$=\mathbb{E}\Big\{\frac{1}{n^2}\sum_{i_1,i_2\in[n]}K(\mathrm{X}_{i_1},\mathrm{Y}_{i_1},\mathrm{X}_{i_2},\mathrm{Y}_{i_2})+\frac{1}{m^2}\sum_{j_1,j_2\in[m]}K(\widetilde{\mathrm{X}}_{j_1},\widetilde{\mathrm{Y}}_{j_1},\widetilde{\mathrm{X}}_{j_2},\widetilde{\mathrm{Y}}_{j_2})$$

$$-\frac{2}{nm}\sum_{i\in[n],j\in[m]}K(\mathrm{X}_i,\mathrm{Y}_i,\widetilde{\mathrm{X}}_j,\widetilde{\mathrm{Y}}_j)\Big\}$$

$$=\Big(\frac{1}{n}+\frac{1}{m}\Big)\mathbb{E}\{K(\mathrm{Z},\mathrm{Z})-K(\mathrm{Z},\mathrm{Z}')\}\leq\Big(\frac{1}{n}+\frac{1}{m}\Big)\kappa^2\qquad(36)$$

where Z and Z′ denote two independent copies of the RV $(X, Y)$.

Therefore, the bound in (26) follows using McDiarmid's inequality. Specifically, with probability at least $1 - \delta$ we get

$$\sup_{f \in \mathcal{F}_1} \left| \frac{1}{n} \sum_{i \in [n]} f(X_i, Y_i) - \frac{1}{m} \sum_{i \in [m]} f(\widetilde{X}_i, \widetilde{Y}_i) \right|$$

$$= \left\| \frac{1}{n} \sum_{i \in [n]} K(X_i, Y_i, \cdot) - \frac{1}{m} \sum_{i \in [m]} K(\widetilde{X}_i, \widetilde{Y}_i, \cdot) \right\|_{\mathcal{H}} \qquad (37)$$

$$\leq \kappa \sqrt{\frac{1}{n} + \frac{1}{m}} + R\kappa \sqrt{\left( \frac{1}{n} + \frac{1}{m} \right) \log \frac{1}{\delta}} \qquad (38)$$

because changing any $(X_i, Y_i)$ or $(\widetilde{X}_i, \widetilde{Y}_i)$ by other sample in (37), changes the norm at most $\sqrt{2}\kappa/n$ or $\sqrt{2}\kappa/m$, as a consequence of the triangular inequality. Moreover, the bound in (26) is satisfied simultaneously for all $R > 0$ with probability at least $1 - \delta$ because

$$\sup_{f \in \mathcal{F}_R} \left| \frac{1}{n} \sum_{i \in [n]} f(X_i, Y_i) - \frac{1}{m} \sum_{i \in [m]} f(\widetilde{X}_i, \widetilde{Y}_i) \right| = R \sup_{f \in \mathcal{F}_1} \left| \frac{1}{n} \sum_{i \in [n]} f(X_i, Y_i) - \frac{1}{m} \sum_{i \in [m]} f(\widetilde{X}_i, \widetilde{Y}_i) \right|$$

for any $R > 0$. $\qquad \square$

## B  Proof of Theorem 1

Let $\widehat{q}$ be the quantile obtained in Step 4 of Algorithm 2. Using (18) in Lemma 1 we have

$$\widehat{q} \geq \text{Quantile}\big(\{(S(X_i, Y_i), 1/n)\}_{i \in [n]}; (1 - \alpha + n\text{E}/(n+1))(1 + 1/n)\big)$$

for

$$\text{E} = \frac{1}{n} \sum_{i \in [n]} \mathfrak{X}_{\mathcal{C}}(X_i, Y_i) - \frac{1}{n} \sum_{i \in [n], y \in \mathcal{Y}} w_i(y) \mathfrak{X}_{\mathcal{C}}(X_i, y).$$

In addition, using (25) in Lemma 2 we have

$$|\text{E}| \leq V_{\text{opt}} + D(\mathfrak{X}_{\mathcal{C}}, \mathcal{F}) + 2(\mathcal{R}_n(\mathcal{F}) + \mathcal{R}_m(\mathcal{F})) + B\sqrt{\left( \frac{1}{n} + \frac{1}{m} \right) \log \frac{\log(2/\delta)}{2}} = G_{n,m}^{\mathcal{F}} \quad (39)$$

with probability at least $1 - \delta/2$. Therefore, with probability at least $1 - \delta/2$ we have

$$\widehat{q} \geq \text{Quantile}\big(\{(S(X_i, Y_i), 1/n)\}_{i \in [n]}; (1 - \alpha - nG_{n,m}^{\mathcal{F}}/(n+1))(1 + 1/n)\big)$$

due to the monotonicity of the quantile function.

Therefore, with probability at least $1 - \delta/2$ we have

$$\mathbb{P}\{Y \in \mathcal{C}(X)|\mathcal{D}\} = \mathbb{P}\{S(X, Y) \leq \widehat{q}|\mathcal{D}\}$$
$$\geq \mathbb{P}\{S(X, Y) \leq \text{Quantile}\big(\{S(X_i, Y_i), 1/n)\}_{i \in [n]}; (1 - \alpha - nG_{n,m}^{\mathcal{F}}/(n+1))(1 + 1/n)\big) |\mathcal{D}\}. \quad (40)$$

Using standard results for training-conditional coverage probabilities in conventional split conformal prediction (see e.g., [2, 10]), in cases with $0 \leq 1 - \alpha - G_{n,m}^{\mathcal{F}} n/(n+1) \leq 1$, the probability in (40) is lower bounded by

$$1 - \alpha - \frac{n}{n+1} G_{n,m}^{\mathcal{F}} - \sqrt{\frac{\log(2/\delta)}{2n}} \geq 1 - \alpha - G_{n,m}^{\mathcal{F}} - \sqrt{\frac{\log(2/\delta)}{2n}} \quad (41)$$

with probability at least $1 - \delta/2$. In addition, the lower bound in (41) is also satisfied in other cases because if $1 - \alpha - G_{n,m}^{\mathcal{F}} n/(n+1) < 0$ the lower bound is trivially valid and the case $1 - \alpha - G_{n,m}^{F} n/(n+1) > 1$ is not possible because both $\alpha$ and $G_{n,m}^{\mathcal{F}}$ are positive. Therefore, the inequality in (8) follows directly from the lower bound in (41) using the union bound.

In case the values $\{S(X_i, y)\}_{(i,y) \in [n] \times \mathcal{Y}}$ are distinct with probability one, using (19) in Lemma 1 we have

$$\widehat{q} \leq \text{Quantile}\big(\{(S(X_i, Y_i), 1/n)\}_{i \in [n]}; (1 - \alpha + nG_{n,m}^{\mathcal{F}}(n+1) + 1/(n+1))(1 + 1/n)\big)$$

with probability at least $1 - \delta/2$.

Therefore, with probability at least $1 - \delta/2$ we have

$$\mathbb{P}\{Y \in \mathcal{C}(X)|\mathcal{D}\} = \mathbb{P}\{S(X, Y) \leq \widehat{q}|\mathcal{D}\}$$

$$\leq \mathbb{P}\{S(X, Y) \leq \text{Quantile}\big(\{S(X_i, Y_i), 1/n)\}_{i \in [n]}; (1 - A)(1 + 1/n)\big)|\mathcal{D}\} \tag{42}$$

with $A = \alpha - nG_{n,m}^{\mathcal{F}}/(n+1) - 1/(n+1)$.

Using standard results for training-conditional coverage probabilities in conventional split conformal prediction (see e.g., [2,10]), in cases with $0 \leq 1-\alpha+G_{n,m}^{\mathcal{F}}n/(n+1)+1/(n+1) \leq 1$, the probability in (42) follows a Beta distribution $\text{Beta}(n + 1 - \lfloor(n + 1)A\rfloor, \lfloor(n + 1)A\rfloor)$ where $\lfloor \cdot \rfloor$ denotes the largest integer that lower bounds the argument. Such a Beta distribution has mean

$$\frac{n + 1 - \lfloor(n+1)A\rfloor}{n + 1} \leq 1 - A + \frac{1}{n + 1} = 1 - \alpha + \frac{2}{n + 1} + \frac{n}{n + 1}G_{n,m}^{\mathcal{F}}$$

and is sub-Gaussian with parameter at most $1/4n$. Hence, using a Chernoff bound the probability in (42) can be upper bounded by

$$1 - \alpha + \frac{2}{n + 1} + \frac{n}{n + 1}G_{n,m}^{\mathcal{F}} + \sqrt{\frac{\log(2/\delta)}{2n}} \leq 1 - \alpha + \frac{2}{n + 1} + G_{n,m}^{\mathcal{F}} + \sqrt{\frac{\log(2/\delta)}{2n}} \tag{43}$$

with probability at least $1 - \delta/2$. In addition, the upper bound in (43) is also satisfied in other cases because if $1 - \alpha + G_{n,m}^{\mathcal{F}}n/(n+1) + 1/(n+1) > 1$ the upper bound is trivially valid and the case $1 - \alpha + G_{n,m}^{\mathcal{F}}n/(n+1) + 1/(n+1) < 0$ is not possible because $\alpha \leq 1$ and $G_{n,m}^{\mathcal{F}} \geq 0$. Therefore, the inequality in (10) follows directly from the upper bound in (43) using the union bound.

## C  Proof of Theorem 2

The proof is analogous to that of Theorem 1 above. The main difference lies in the high-probability bound of $|E|$ for

$$E = \frac{1}{n}\sum_{i \in [n]} \mathfrak{X}_{\mathcal{C}}(X_i, Y_i) - \frac{1}{n}\sum_{i \in [n], y \in \mathcal{Y}} w_i(y)\mathfrak{X}_{\mathcal{C}}(X_i, y).$$

If $\mathcal{F}_R = \{f \in \mathcal{H} : \|f\|_{\mathcal{H}} \leq R\}$, using (26) in Lemma 2, we have

$$|E| \leq D(\mathfrak{X}_{\mathcal{C}}, \mathcal{F}_R) + 2\kappa R\sqrt{\frac{1}{n} + \frac{1}{m}}\Big(1 + \sqrt{\log(2/\delta)}\Big), \text{ for all } R > 0$$

with probability at least $1 - \delta/2$. Therefore,

$$|E| \leq \inf_{R > 0} D(\mathfrak{X}_{\mathcal{C}}, \mathcal{F}_R) + 2\kappa R\sqrt{\frac{1}{n} + \frac{1}{m}}\Big(1 + \sqrt{\log(2/\delta)}\Big)$$

$$= \min_{f \in \mathcal{H}} \frac{1}{n}\sum_{i \in [n], y \in \mathcal{Y}} \big|\mathfrak{X}_{\mathcal{C}}(X_i, y) - f(X_i, y)\big| + 2\kappa\big(1 + \sqrt{\log(2/\delta)}\big)\sqrt{\frac{1}{n} + \frac{1}{m}}\|f\|_{\mathcal{H}} = G_{n,m}^{K}$$

$$\tag{44}$$

with probability at least $1 - \delta/2$.

The rest of the proof is totally analogous to that for Theorem 1 (see Appendix B for the details). In particular, the bound for $|E|$ in (44) results in

$$\widehat{q} \geq \text{Quantile}\big(\{(S(X_i, Y_i), 1/n)\}_{i \in [n]}; (1 - \alpha - nG_{n,m}^{K}/(n+1))(1 + 1/n)\big), \text{ w.p. } \geq 1 - \frac{\delta}{2}$$

using (18) in Lemma 1, and

$$\widehat{q} \leq \text{Quantile}\big(\{(S(X_i, Y_i), 1/n)\}_{i \in [n]}; (1 - \alpha + nG_{n,m}^{K}(n+1) + 1/(n+1))(1 + 1/n)\big), \text{ w.p. } \geq 1 - \frac{\delta}{2}$$

if $\{S(X_i, y)\}_{(i,y) \in [n] \times \mathcal{Y}}$ are distinct with probability one, using (19) in Lemma 1.

Therefore, (13) and (15) are obtained taking into account that

$$\mathbb{P}\{Y \in \mathcal{C}(X)|\mathcal{D}\} = \mathbb{P}\{S(X, Y) \leq \widehat{q}|\mathcal{D}\}$$

and that, for any $0 \leq \alpha' \leq 1$, the RV

$$\mathbb{P}\{S(X, Y) \leq \text{Quantile}\big(\{(S(X_i, Y_i), 1/n)\}_{i \in [n]}; (1 - \alpha')(1 + 1/n)\big)|\mathcal{D}\}$$

follows a distribution that is dominated by a $\text{Beta}(n + 1 - \lfloor(n+1)\alpha'\rfloor, \lfloor(n+1)\alpha'\rfloor)$, and is equal to such a Beta distribution in the case where $\{S(X_i, y)\}_{(i,y) \in [n] \times \mathcal{Y}}$ are distinct with probability one.

## D  Additional results

The following provides the detailed derivations leading to (11) and (12) in the main paper.

**Proposition 1.** Let $\varepsilon_{\text{opt}}$ be an upper bound for the difference between $\Phi_{\mathcal{F}}(\mathbf{w})$ and the optimum value of (4). If the feasible set in (4) includes the label weights $\mathbf{w}^*$ corresponding to the actual calibration labels, with probability at least $1 - \delta$ we have

$$\Phi_{n,m}^{\mathcal{F}}(\mathbf{w}) \leq \varepsilon_{\text{opt}} + 2(\mathcal{R}_n(\mathcal{F}) + R_m(\mathcal{F})) + B\sqrt{\frac{1}{2}\Big(\frac{1}{n} + \frac{1}{m}\Big)\log\frac{1}{\delta}}. \tag{45}$$

*Proof.* If $\Phi^*$ denotes the optimum value of (4), we have

$$\Phi_{n,m}^{\mathcal{F}}(\mathbf{w}) = (\Phi_{n,m}^{\mathcal{F}}(\mathbf{w}) - \Phi^*) + \Phi^* \leq \varepsilon_{\text{opt}} + \Phi^* \tag{46}$$

and since $\mathbf{w}^*$ is included in the feasible set of (4), we have

$$\Phi^* \leq \sup_{f\in\mathcal{F}} \Big| \frac{1}{n}\sum_{i\in[n]} f(X_i, Y_i) - \frac{1}{m}\sum_{i\in[m]} f(\widetilde{X}_i, \widetilde{Y}_i) \Big|. \tag{47}$$

Using McDiarmid's inequality (see e.g., [16]) with probability at least $1 - \delta$ we have

$$\sup_{f\in\mathcal{F}} \Big| \frac{1}{n}\sum_{i\in[n]} f(X_i, Y_i) - \frac{1}{m}\sum_{i\in[m]} f(\widetilde{X}_i, \widetilde{Y}_i) \Big| \leq 2(\mathcal{R}_n(\mathcal{F}) + \mathcal{R}_m(\mathcal{F})) + B\sqrt{\frac{1}{2}\Big(\frac{1}{n} + \frac{1}{m}\Big)\log\frac{1}{\delta}} \tag{48}$$

because changing any $(X_i, Y_i)$ or $(\widetilde{X}_i, \widetilde{Y}_i)$ by other sample in the left hand side of (48), changes the 'sup' at most $B/n$ or $B/m$, and we have

$$\mathbb{E}\Big\{ \sup_{f\in\mathcal{F}} \Big| \frac{1}{n}\sum_{i\in[n]} f(X_i, Y_i) - \frac{1}{m}\sum_{i\in[m]} f(\widetilde{X}_i, \widetilde{Y}_i) \Big| \Big\} \leq 2(\mathcal{R}_n(\mathcal{F}) + \mathcal{R}_m(\mathcal{F})) \tag{49}$$

using a symmetrization argument, as in the proof of Lemma 2.

Therefore, the result in (45) is obtained by combining the inequalities in (46), (47), and (48). $\qquad\square$

**Proposition 2.** Let $\mathcal{H}$ be an RKHS given by kernel $K$ over $\mathcal{Z} = \mathcal{X} \times \mathcal{Y}$. If $\mathcal{F}$ is the family of real-valued functions $\mathcal{F} = \{f \in \mathcal{H} : \|f\|_{\mathcal{H}} \leq 1\}$, we have

$$\sup_{f\in\mathcal{F}} \Big| \frac{1}{n}\sum_{(i,y)\in[n]\times\mathcal{Y}} w_i(y)f(X_i, y) - \frac{1}{m}\sum_{i\in[m]} f(\widetilde{X}_i, \widetilde{Y}_i) \Big| = \Big( \frac{1}{n^2}\mathbf{w}^\top\mathbf{K}\mathbf{w} - \frac{2}{nm}\mathbf{v}^\top\mathbf{w} + \frac{1}{m^2}\mathbf{1}^\top\widetilde{\mathbf{K}}\mathbf{1} \Big)^{\frac{1}{2}}. \tag{50}$$

*Proof.* Using the reproducing property of RKHSs and the dual characterization of the norm in $\mathcal{H}$, we have

$$\sup_{f:\|f\|_{\mathcal{H}}\leq 1} \Big| \frac{1}{n}\sum_{(i,y)\in[n]\times\mathcal{Y}} w_i(y)f(X_i, y) - \frac{1}{m}\sum_{i\in[m]} f(\widetilde{X}_i, \widetilde{Y}_i) \Big|^2$$

$$= \sup_{f:\|f\|_{\mathcal{H}}\leq 1} \Big| \frac{1}{n}\sum_{(i,y)\in[n]\times\mathcal{Y}} w_i(y) <f, K((X_i, y), \cdot)>_{\mathcal{H}} - \frac{1}{m}\sum_{i\in[m]} <f, K((\widetilde{X}_i, \widetilde{Y}_i), \cdot)>_{\mathcal{H}} \Big|^2$$

$$= \sup_{f:\|f\|_{\mathcal{H}}\leq 1} \Big| <f, \frac{1}{n}\sum_{(i,y)\in[n]\times\mathcal{Y}} w_i(y)K((X_i, y), \cdot)>_{\mathcal{H}} - <f, \frac{1}{m}\sum_{i\in[m]} K((\widetilde{X}_i, \widetilde{Y}_i), \cdot)>_{\mathcal{H}} \Big|^2 \tag{51}$$

$$
\begin{aligned}
&= \sup_{f:\|f\|_{\mathcal{H}} \leq 1} \left| < f, \frac{1}{n} \sum_{(i,y)\in[n]\times\mathcal{Y}} w_i(y)K((X_i,y),\cdot) - \frac{1}{m} \sum_{i\in[m]} K((\widetilde{X}_i,\widetilde{Y}_i),\cdot) >_{\mathcal{H}} \right|^2 \\
&= \left\| \frac{1}{n} \sum_{(i,y)\in[n]\times\mathcal{Y}} w_i(y)K((X_i,y),\cdot) - \frac{1}{m} \sum_{i\in[m]} K((\widetilde{X}_i,\widetilde{Y}_i),\cdot) \right\|_{\mathcal{H}}^2 \\
&= \frac{1}{n^2} \sum_{(i_1,y_1),(i_2,y_2)\in[n]\times\mathcal{Y}} w_{i_1}(y_1)w_{i_2}(y_2)K((X_{i_1},y_1),(X_{i_2},y_2)) \\
&\quad - \frac{2}{nm} \sum_{(i_1,y_1)\in[n]\times\mathcal{Y},i_2\in[m]} w_{i_1}(y_1)K((X_{i_1},y_1),(\widetilde{X}_{i_2},\widetilde{Y}_j)) \\
&\quad + \frac{1}{m^2} \sum_{i_1,i_2\in[m]} K((\widetilde{X}_{i_1},\widetilde{Y}_{i_1}),(\widetilde{X}_{i_2},\widetilde{Y}_{i_2}))
\end{aligned}
\tag{52}
$$

where $< \cdot, \cdot >_{\mathcal{H}}$ denotes the inner product in $\mathcal{H}$. Then, the result in (50) is directly obtained using the definition of $\mathbf{K} \in \mathbb{R}^{nc\times nc}$, $\widetilde{\mathbf{K}} \in \mathbb{R}^{m\times m}$, and $\mathbf{v} \in \mathbb{R}^{nc}$ as

$$
K_{i,j} = K(Z_i,Z_j), \text{ for } i,j\in[nc], \ \widetilde{K}_{i,j} = K(\widetilde{Z}_i,\widetilde{Z}_j), \text{ for } i,j\in[m]
$$
$$
v_i = \sum_{j\in[m]} K(Z_i,\widetilde{Z}_j), \text{ for } i\in[nc]
$$

for $Z_{(i-1)c+y} = (X_i,y)$ for $i\in[n], y\in\mathcal{Y}$, and $\widetilde{Z}_i = (\widetilde{X}_i,\widetilde{Y}_i)$ for $i\in[m]$. $\qquad\square$

# E   Implementation details and additional experimental results

In the following we provide further implementation details and describe the datasets used in the experimental results. Then, we complement the results in the main paper by includ-ing the results for the running times of the methods presented as well as additional results with other target coverages and conformal scores. In addition, the Github `https://github.com/MachineLearningBCAM/Unsupervised-conformal-prediction-NeurIPS2025` and the folder Implementation_Unsupervised_Conformal in the supplementary materials provide Matlab code corresponding to the presented methods with the setting used in the numerical results.

## E.1   Implementation and datasets details

We utilize 9 publicly available and common benchmark datasets for classification tasks: 'Drybean', 'Forestcov', 'Satellite', 'USPS', 'MNIST', 'FashionMNIST', 'CIFAR10', 'ImageNet10' (first 10 classes of ImageNet), and 'Letter'. These datasets can be found in the UCI repository [23], Tensor Flow datasets [24], and Kaggle website `https://www.kaggle.com`. The main characteristics of the datasets used is provided in Table 2.

Table 2: Key characteristics of benchmark datasets used in the experiments.

| Dataset | # Samples | # Classes | Input Dimensionality |
|---|---|---|---|
| Drybean | 13,611 | 7 | 16 |
| Forestcov | 581,012 | 7 | 54 |
| Satellite | 6,435 | 6 | 36 |
| Letter | 20,000 | 26 | 16 |
| USPS | 9,298 | 10 | 256 |
| MNIST | 70,000 | 10 | 784 |
| FashionMNIST | 70,000 | 10 | 784 |
| CIFAR10 | 60,000 | 10 | 3,072 |
| ImageNet10 | 13,000 | 10 | 150,528 |

In each random realization, the datasets are randomly partitioned in training, calibration, and test sets. The sizes of the training and test sets are $3,000$ samples and $1,000$, and that of the calibra-tion set is varied from $10$ to $3,000$. The classification rules are obtained using random forests in the tabular datasets ('Drybean', 'Forestcov', 'Satellite', and 'Letter') and using neural networks in

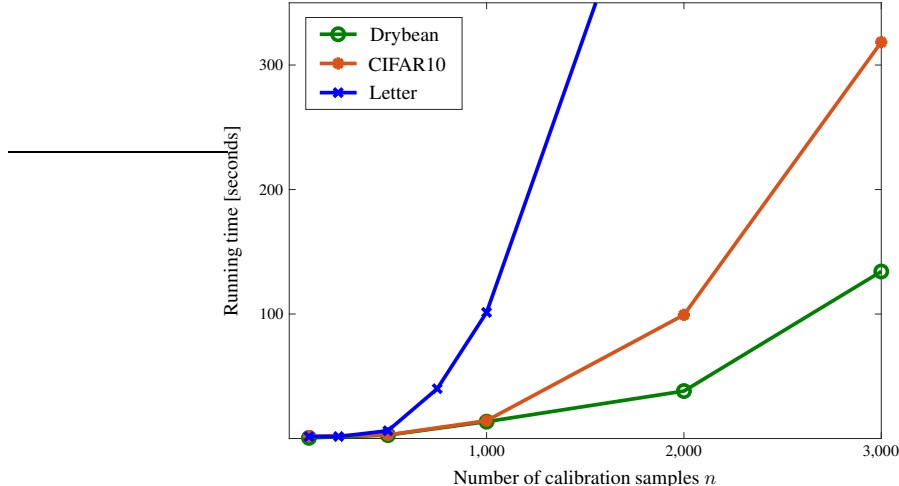

Figure 3: Running times of the methods presented as the number of calibration samples increases. For datasets with up to 10 classes, the running times range from tens to hundreds of seconds using few thousands of samples.

the image datasets ('USPS', 'MNIST', 'FashionMNIST', 'CIFAR10', and 'ImageNet10'). Specifically, the random forests are given by 200 decision trees and learning is carried out with 20 maximum number of splits and 10 minimum leaf size. The neural networks for 'USPS', 'MNIST', and 'FashionMNIST' datasets have two hidden layers of sizes 128 and 64 and learning is carried out with regularization parameter 0.001. The implementation for 'CIFAR10' and 'ImageNet10' datasets is slightly different due their higher complexity. In particular, the classification rule for those datasets is learned only once by fine-tuning a Resnet50 until a validation error of 4% in 'CIFAR10' and 11% in 'ImageNet10' datasets, using SGD with momentum, initial learning rate of 0.001, minibatch size of 64, and regularization parameter 0.001.

Set-prediction rules are obtained using the adaptive conformal score proposed in [2,9] and also using the conformal score directly given by probability estimates [1]. Such scores are computed from the probability estimates provided by the classification rules (proportion of votes from trees in random forest methods, and soft-max outputs in neural networks methods). The conventional approach for split conformal prediction with supervised calibration is implemented using Algorithm 1 (see also, e.g., [2]). The naive approach with unsupervised calibration is implemented using Algorithm 1 with labels corresponding to the predictions obtained by the classification rules.

The presented method is implemented using Algorithm 2 with label weights obtained by Algorithm 3 using a Gaussian kernel selected from a set given by 10 candidate bandwidth (scale) parameters $\sigma = \sigma_0 \sqrt{d/2}$, for $\sigma_0 \in \{10^{-1+t/3}, t = 0, 1, \ldots, 9\}$, and $m = n$ training samples randomly drawn from the samples used to learn the classification rule. The inequality constraints in (12) given by $\mathbf{B}$ and $\mathbf{b}$ correspond to estimates for the cross-entropy loss of the classification rule obtained in the training stage. Specifically, $\mathbf{B} \in \mathbb{R}^{1 \times nc}$ is given as in (5) and $\mathbf{b} = nL \in \mathbb{R}$ with $L$ the estimate of the cross-entropy plus its sample standard deviation obtained using validation samples (10-fold cross validation in all datasets but in 'CIFAR10' and 'ImageNet10' datasets for which 1,000 validation samples were used for those estimates). In addition, for 'CIFAR10' and 'ImageNet10' datasets the instances $\{\widetilde{X}_i\}_{i \in [m]}$ and $\{X_i\}_{i \in [m]}$ used in Algorithm 3 are given by the 512-dimensional feature representations given by the penultimate layer of the pre-trained Resnet18 network.

### E.2 Running times of the methods presented

The main computational burden of the methods presented corresponds to the Step 11 in Algorithm 3 that solves a quadratic optimization problem. In the experimental results carried out, such optimization problem is solved using interior point methods with Mosek solver `https://www.mosek.com`. Figure 3 shows the running time achieved by Algorithm 3 using a regular desktop machine in the datasets 'Drybean', 'CIFAR10', and 'Letter'. These datasets are representative of the overall running time because they have assorted characteristics. In particular the number of classes in those datasets is 7 ('Drybean'), 10 ('CIFAR10'), and 26 ('Letter'). As shown in the figure, the pre-

sented methods have running times ranging from tens to hundreds of seconds for cases with up to 10 classes and up to 3,000 calibration samples. In line with the discussion in Section 4 of the paper, the only problematic cases are those composed by a sizable number of classes since the computational complexity of the methods presented increases cubically with the number of classes.

### E.3 Additional results with the adaptive conformal score

The main paper shows experimental results using the adaptive conformal score [2, 9] given by

$$S(X, Y) = \sum_{i \in [c]} \widehat{p}(i|X) \mathbb{I}\{\widehat{p}(i|X) > \widehat{p}(Y|X)\} + \mathrm{U}\,\widehat{p}(Y|X)$$

for U a random value from $\mathrm{Unif}(0, 1)$ and $\widehat{p}$ the probability estimate provided by the classification rule used. This section complements the results in the main paper with other target probabilities and with other datasets for figures such as those in Fig. 1 and Fig. 2. Specifically, Tables 3 and 4 show coverage probabilities and prediction set sizes analogous to those in Table 1 in the main paper but using target probabilities $1 - \alpha = 0.95$ and $1 - \alpha = 0.85$. In addition, Figures 4, 5, and 6 show violin plots such as those in Fig. 1 using 'ImageNet10', 'FashionMNIST', and datasets, and Figure 7 shows the decrease of coverage gap with the number of calibration samples using 'ImageNet10' and 'MNIST' datasets.

Table 3: Coverage probability and prediction set size for conventional approach with supervised calibration ('Supervised'), proposed approach with unsupervised calibration ('Unsupervised'), and naive unsupervised approach ('Unsup. Naive'). Results are shown as: mean (interquartile interval), and correspond to $1 - \alpha = .95$.

| Dataset | Coverage probability | | | Prediction set size | |
|---|---|---|---|---|---|
| | Supervised | Unsupervised | Unsup. Naive | Supervised | Unsupervised |
| Drybean | 0.95 (0.94,0.96) | 0.95 (0.94,0.96) | 0.91 (0.91,0.92) | 1.52 (1.47,1.56) | 1.51 (1.47,1.55) |
| Forestcov | 0.95 (0.94,0.96) | 0.94 (0.93,0.95) | 0.68 (0.67,0.69) | 2.24 (2.18,2.29) | 2.15 (2.11,2.19) |
| Satellite | 0.95 (0.94,0.96) | 0.95 (0.94,0.96) | 0.91 (0.90,0.92) | 1.77 (1.71,1.82) | 1.79 (1.73,1.84) |
| USPS | 0.95 (0.94,0.96) | 0.95 (0.95,0.96) | 0.94 (0.93,0.94) | 1.32 (1.28,1.35) | 1.33 (1.29,1.36) |
| MNIST | 0.95 (0.94,0.95) | 0.95 (0.95,0.96) | 0.93 (0.92,0.93) | 1.43 (1.38,1.48) | 1.45 (1.41,1.50) |
| Fashion | 0.95 (0.94,0.96) | 0.95 (0.95,0.96) | 0.87 (0.86,0.88) | 1.84 (1.76,1.90) | 1.84 (1.79,1.89) |
| CIFAR10 | 0.95 (0.94,0.96) | 0.95 (0.95,0.96) | 0.93 (0.93,0.94) | 1.10 (1.08,1.12) | 1.11 (1.10,1.13) |
| ImagNet10 | 0.95 (0.94,0.96) | 0.94 (0.94,0.95) | 0.93 (0.92,0.94) | 2.20 (2.13,2.27) | 2.07 (2.00,2.14) |
| Letter | 0.95 (0.94,0.96) | 0.90 (0.89,0.92) | 0.80 (0.79,0.82) | 9.06 (8.38,9.64) | 5.63 (4.99,6.33) |

Table 4: Coverage probability and prediction set size for conventional approach with supervised calibration ('Supervised'), proposed approach with unsupervised calibration ('Unsupervised'), and naive unsupervised approach ('Unsup. Naive'). Results are shown as: mean (interquartile interval), and correspond to $1 - \alpha = .85$.

| Dataset | Coverage probability | | | Prediction set size | |
|---|---|---|---|---|---|
| | Supervised | Unsupervised | Unsup. Naive | Supervised | Unsupervised |
| Drybean | 0.85 (0.84,0.86) | 0.86 (0.85,0.87) | 0.78 (0.76,0.79) | 1.09 (1.06,1.11) | 1.11 (1.09,1.13) |
| Forestcov | 0.85 (0.84,0.86) | 0.86 (0.85,0.87) | 0.57 (0.56,0.58) | 1.65 (1.61,1.69) | 1.70 (1.67,1.72) |
| Satellite | 0.85 (0.84,0.86) | 0.85 (0.84,0.86) | 0.78 (0.76,0.79) | 1.27 (1.24,1.30) | 1.26 (1.23,1.29) |
| USPS | 0.85 (0.84,0.86) | 0.85 (0.84,0.86) | 0.83 (0.81,0.84) | 1.00 (0.97,1.01) | 1.01 (0.98,1.03) |
| MNIST | 0.85 (0.84,0.86) | 0.86 (0.84,0.87) | 0.81 (0.80,0.82) | 1.06 (1.03,1.08) | 1.07 (1.05,1.09) |
| Fashion | 0.85 (0.84,0.86) | 0.86 (0.85,0.87) | 0.74 (0.72,0.75) | 1.24 (1.21,1.27) | 1.26 (1.24,1.29) |
| CIFAR10 | 0.85 (0.84,0.86) | 0.86 (0.85,0.87) | 0.82 (0.81,0.83) | 0.93 (0.91,0.94) | 0.94 (0.92,0.95) |
| ImagNet10 | 0.85 (0.84,0.86) | 0.85 (0.84,0.87) | 0.80 (0.79,0.81) | 1.37 (1.32,1.41) | 1.38 (1.33,1.43) |
| Letter | 0.85 (0.84,0.86) | 0.81 (0.80,0.83) | 0.60 (0.58,0.62) | 4.07 (3.84,4.28) | 3.38 (3.18,3.58) |

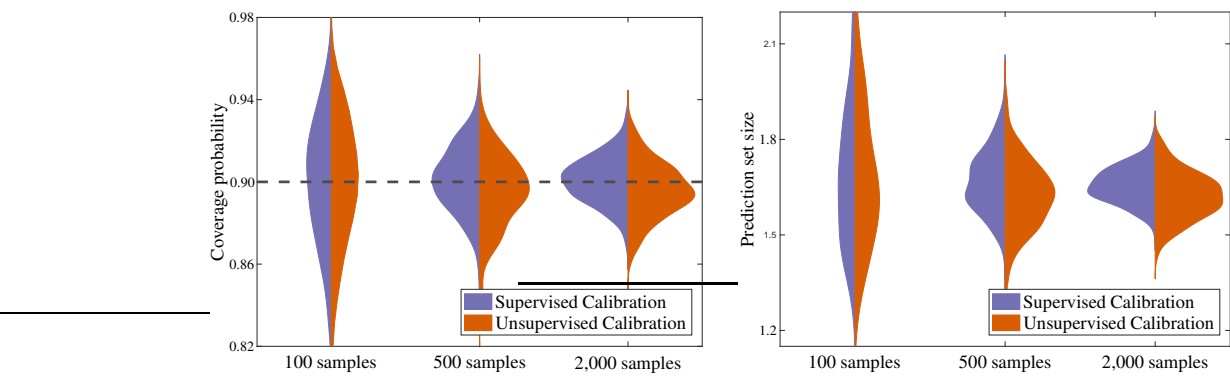

Figure 4: Coverage probabilities and prediction set sizes over $400$ random partitions of 'ImageNet10' dataset with target coverage $1 - \alpha = 0.9$ and number of calibration samples ranging from $100$ to $2,000$.

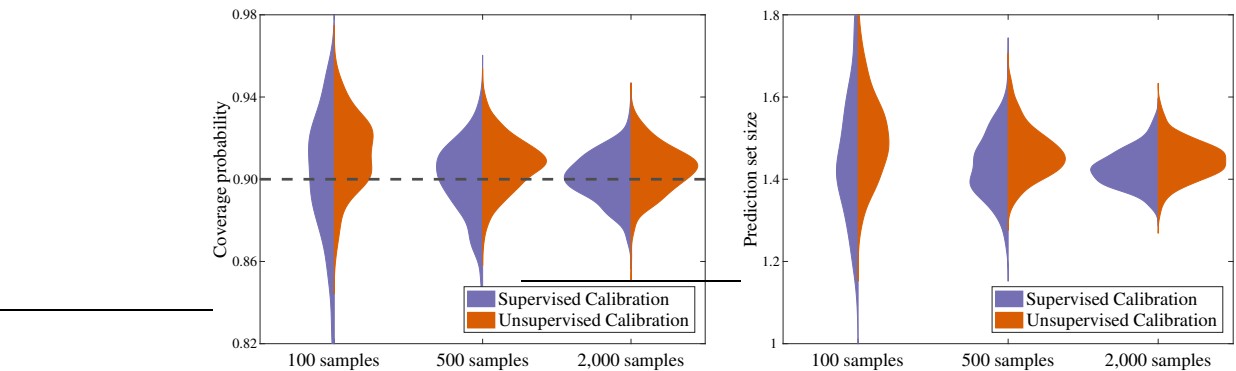

Figure 5: Coverage probabilities and prediction set sizes over $400$ random partitions of 'FashionMNIST' dataset with target coverage $1 - \alpha = 0.9$ and number of calibration samples ranging from $100$ to $2,000$.

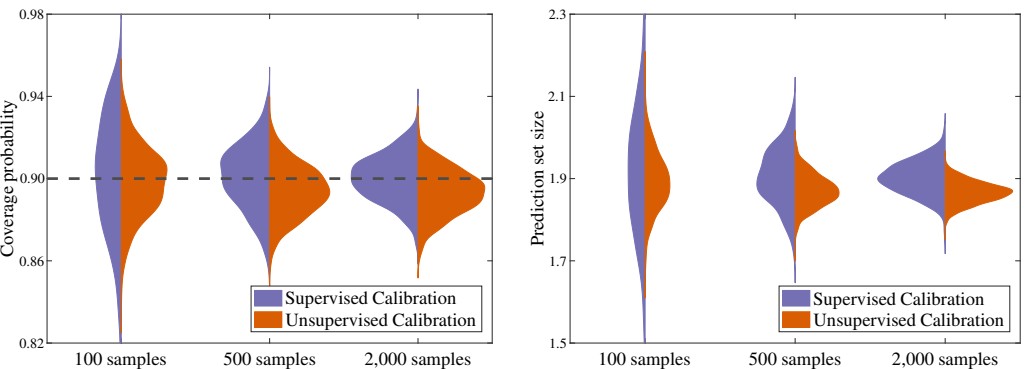

Figure 6: Coverage probabilities and prediction set sizes over $400$ random partitions of 'Forestcov' dataset with target coverage $1 - \alpha = 0.9$ and number of calibration samples ranging from $100$ to $2,000$.

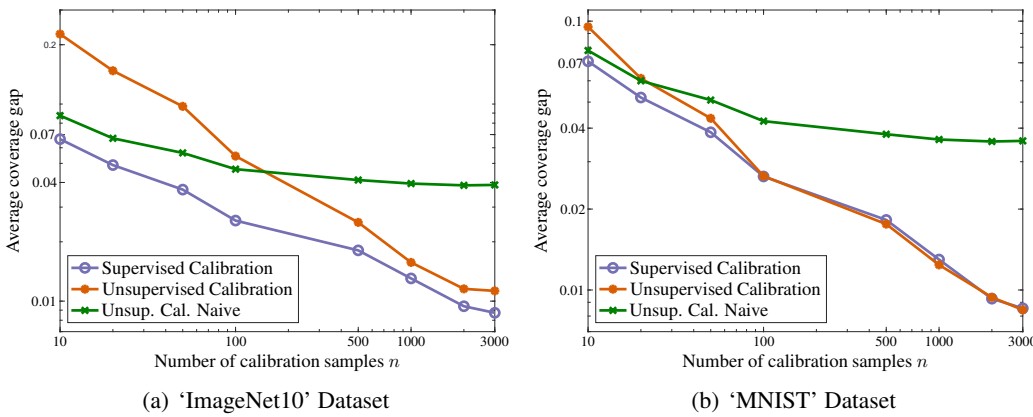

(a) 'ImageNet10' Dataset  (b) 'MNIST' Dataset

Figure 7: Decrease of average coverage gap with the number of calibration samples.

## E.4   Additional results with the conformal score given by probability estimates

This section complements the results provided above by using the conformal score given by probability estimates, i.e., $S(X, Y) = 1 - \widehat{p}(Y|X)$ for $\widehat{p}$ the probability estimate provided by the classification rule used. Specifically, Tables 5, 6, and 7 show coverage probabilities and prediction set for target coverages $1 - \alpha \in \{0.9, 0.95, 0.85\}$, Figures 8, 9, and 10 show violin plots using 'CIFAR10', 'FashionMNIST', and 'Forestcov' datasets, and Figures 11 and 11 show decrease of coverage gap with the number of calibration samples using 'CIFAR10', 'Drybean', 'FashionMNIST', and 'MNIST' datasets.

Table 5: Coverage probability and prediction set size for conventional approach with supervised calibration ('Supervised'), proposed approach with unsupervised calibration ('Unsupervised'), and naive unsupervised approach ('Unsup. Naive'). Results are shown as: mean (interquartile interval), and correspond to the conformal score given by probability estimates with $1 - \alpha = 0.9$.

| Dataset | Coverage probability | | | Prediction set size | |
|---|---|---|---|---|---|
| | Supervised | Unsupervised | Unsup. Naive | Supervised | Unsupervised |
| Drybean | 0.90 (0.89,0.91) | 0.89 (0.89,0.90) | 0.84 (0.83,0.85) | 1.02 (1.01,1.04) | 1.01 (1.00,1.02) |
| Forestcov | 0.90 (0.89,0.91) | 0.90 (0.89,0.91) | 0.64 (0.63,0.65) | 1.74 (1.69,1.79) | 1.75 (1.72,1.77) |
| Satellite | 0.90 (0.89,0.91) | 0.90 (0.89,0.91) | 0.84 (0.83,0.86) | 1.44 (1.40,1.48) | 1.44 (1.41,1.48) |
| USPS | 0.90 (0.89,0.91) | 0.91 (0.90,0.92) | 0.89 (0.88,0.90) | 0.91 (0.90,0.92) | 0.93 (0.92,0.94) |
| MNIST | 0.90 (0.89,0.91) | 0.92 (0.91,0.92) | 0.87 (0.86,0.88) | 0.94 (0.93,0.95) | 0.97 (0.96,0.98) |
| Fashion | 0.90 (0.89,0.91) | 0.91 (0.90,0.92) | 0.79 (0.78,0.80) | 1.19 (1.16,1.22) | 1.24 (1.23,1.26) |
| CIFAR10 | 0.90 (0.89,0.91) | 0.91 (0.90,0.92) | 0.89 (0.88,0.90) | 0.91 (0.90,0.92) | 0.92 (0.91,0.93) |
| Letter | 0.90 (0.89,0.91) | 0.86 (0.85,0.87) | 0.72 (0.70,0.73) | 5.22 (4.85,5.56) | 3.91 (3.72,4.09) |

Table 6: Coverage probability and prediction set size for conventional approach with supervised calibration ('Supervised'), proposed approach with unsupervised calibration ('Unsupervised'), and naive unsupervised approach ('Unsup. Naive'). Results are shown as: mean (interquartile interval), and correspond to the conformal score given by probability estimates with $1 - \alpha = 0.95$.

| Dataset | Coverage probability | | | Prediction set size | |
|---|---|---|---|---|---|
| | Supervised | Unsupervised | Unsup. Naive | Supervised | Unsupervised |
| Drybean | 0.95 (0.94,0.96) | 0.95 (0.94,0.96) | 0.86 (0.86,0.87) | 1.16 (1.13,1.18) | 1.16 (1.14,1.18) |
| Forestcov | 0.95 (0.94,0.96) | 0.94 (0.93,0.94) | 0.67 (0.66,0.68) | 2.10 (2.06,2.14) | 2.00 (1.98,2.02) |
| Satellite | 0.95 (0.94,0.96) | 0.95 (0.94,0.96) | 0.91 (0.90,0.92) | 1.78 (1.72,1.83) | 1.79 (1.71,1.84) |
| USPS | 0.95 (0.94,0.96) | 0.96 (0.95,0.97) | 0.93 (0.92,0.93) | 0.99 (0.98,1.00) | 1.01 (1.00,1.02) |
| MNIST | 0.95 (0.94,0.96) | 0.96 (0.95,0.96) | 0.90 (0.90,0.91) | 1.07 (1.05,1.09) | 1.11 (1.09,1.13) |
| Fashion | 0.95 (0.94,0.96) | 0.95 (0.94,0.96) | 0.82 (0.81,0.82) | 1.55 (1.49,1.63) | 1.56 (1.53,1.60) |
| CIFAR10 | 0.95 (0.94,0.96) | 0.96 (0.96,0.97) | 0.93 (0.92,0.94) | 0.98 (0.98,0.99) | 1.01 (1.00,1.02) |
| Letter | 0.95 (0.94,0.96) | 0.90 (0.90,0.91) | 0.75 (0.74,0.76) | 8.66 (8.18,9.12) | 5.46 (5.18,5.72) |

Table 7: Coverage probability and prediction set size for conventional approach with supervised calibration ('Supervised'), proposed approach with unsupervised calibration ('Unsupervised'), and naive unsupervised approach ('Unsup. Naive'). Results are shown as: mean (interquartile interval), and correspond to the conformal score given by probability estimates with $1 - \alpha = 0.85$.

| Dataset | Coverage probability | | | Prediction set size | |
| | Supervised | Unsupervised | Unsup. Naive | Supervised | Unsupervised |
|---|---|---|---|---|---|
| Drybean | 0.85 (0.84,0.86) | 0.85 (0.84,0.86) | 0.81 (0.80,0.82) | 0.92 (0.90,0.93) | 0.92 (0.91,0.93) |
| Forestcov | 0.85 (0.84,0.86) | 0.86 (0.85,0.87) | 0.62 (0.61,0.63) | 1.49 (1.46,1.52) | 1.54 (1.51,1.56) |
| Satellite | 0.85 (0.84,0.86) | 0.86 (0.85,0.87) | 0.62 (0.61,0.63) | 1.49 (1.46,1.52) | 1.54 (1.51,1.56) |
| USPS | 0.85 (0.84,0.86) | 0.86 (0.85,0.87) | 0.84 (0.83,0.85) | 0.86 (0.85,0.87) | 0.87 (0.86,0.88) |
| MNIST | 0.85 (0.84,0.86) | 0.86 (0.85,0.87) | 0.83 (0.82,0.84) | 0.87 (0.86,0.88) | 0.89 (0.88,0.90) |
| Fashion | 0.85 (0.84,0.86) | 0.87 (0.86,0.88) | 0.76 (0.75,0.78) | 1.02 (1.00,1.04) | 1.08 (1.06,1.09) |
| CIFAR10 | 0.85 (0.84,0.86) | 0.85 (0.84,0.86) | 0.84 (0.83,0.85) | 0.86 (0.84,0.87) | 0.86 (0.85,0.87) |
| Letter | 0.85 (0.84,0.86) | 0.82 (0.81,0.83) | 0.69 (0.68,0.70) | 3.59 (3.34,3.82) | 3.04 (2.92,3.17) |

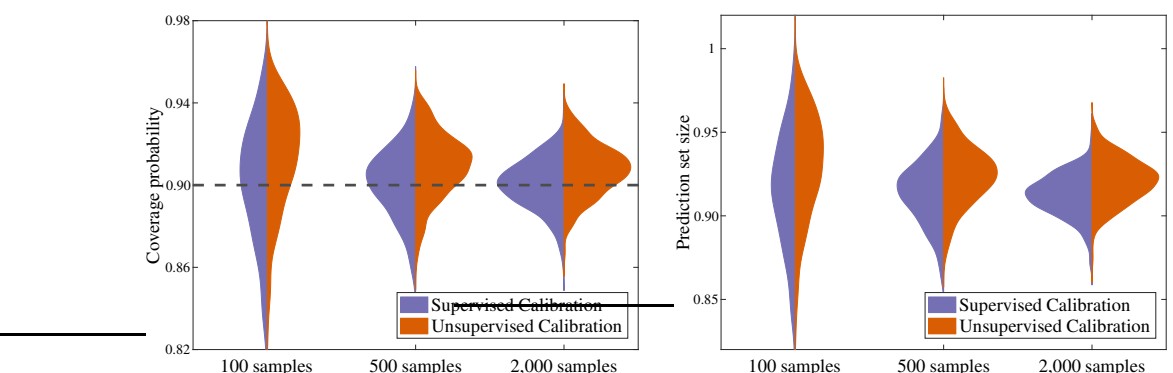

Figure 8: Coverage probabilities and prediction set sizes over 400 random partitions of 'CIFAR10' dataset with conformal score given be probability estimates, target coverage $1-\alpha = 0.9$ and number of calibration samples ranging from 100 to 2,000.

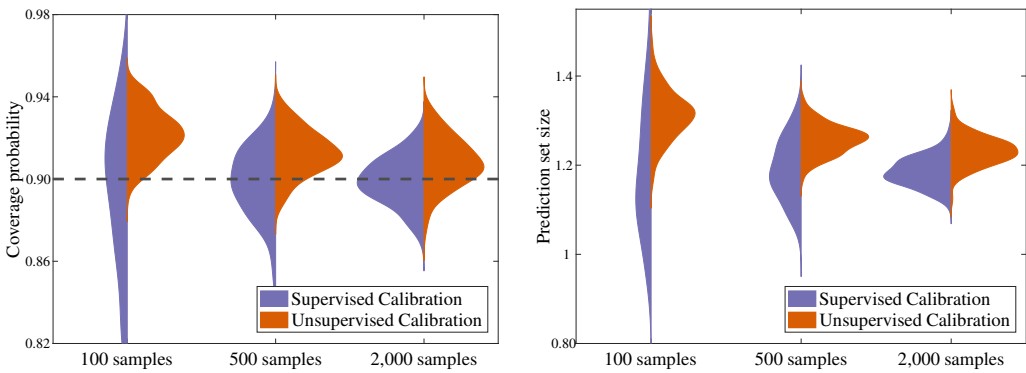

Figure 9: Coverage probabilities and prediction set sizes over 400 random partitions of 'FashionMNIST' dataset with conformal score given be probability estimates, target coverage $1 - \alpha = 0.9$ and number of calibration samples ranging from 100 to 2,000.

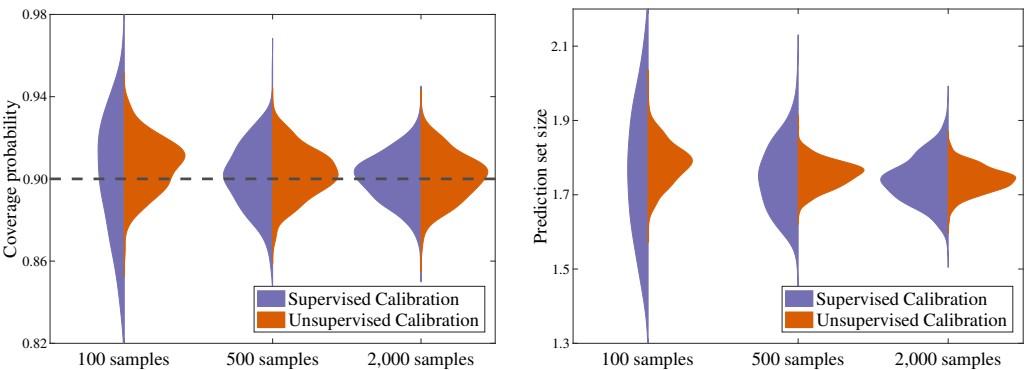

Figure 10: Coverage probabilities and prediction set sizes over 400 random partitions of 'Forestcov' dataset with conformal score given be probability estimates, target coverage $1 - \alpha = 0.9$ and number of calibration samples ranging from 100 to $2,000$.

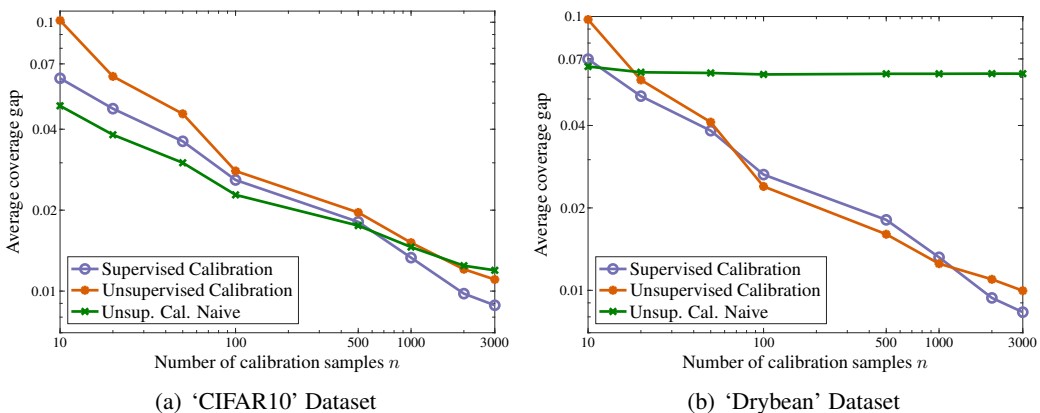

(a) 'CIFAR10' Dataset

(b) 'Drybean' Dataset

Figure 11: Decrease of average coverage gap with the number of calibration samples with conformal score given by probability estimates.

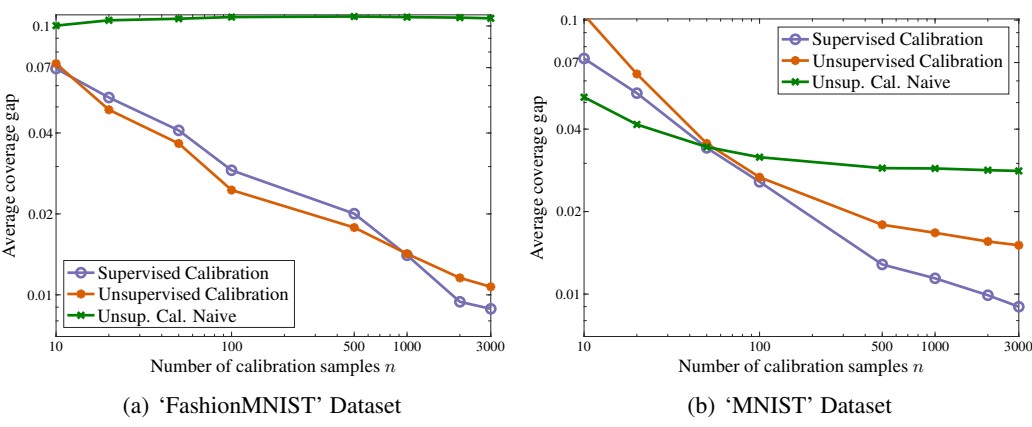

(a) 'FashionMNIST' Dataset

(b) 'MNIST' Dataset

Figure 12: Decrease of average coverage gap with the number of calibration samples with conformal score given by probability estimates.

