# OpenReview forum: "Split conformal classification with unsupervised calibration"
_NeurIPS.cc/2025/Conference — NeurIPS 2025 poster_

### Official Review · Reviewer_cxnj · 2025-06-30

**Clarity:** 2
**Significance:** 3
**Originality:** 3
**Rating:** 5
**Confidence:** 4

**Summary:**

This paper proposes an interesting new idea for using unlabelled calibrated examples for doing split CP. This comes with a worse coverage guarantee, but gives the benefit of using unlabelled data.

**Questions:**

1)
This paper claims that the theoretical guarantee (validity) is "comparable" to that of split CP. However, there is no understanding of what the $G^F_{n,m}$ means in practical terms. What is its magnitude? Can you estimate its value?
I fully appreciate the discussion provided in 3.2 surrounding this, but it's still very hard to understand what the validity guarantee means in practical terms.
I think the claim that the proposal is comparable to split CP is unsubstantiated unless concrete numbers (or a more interpretable expression for this term) is given.

2)
The method is only compared against split CP. However, there's numerous methods which try to achieve the same (i.e., to maximize the information available to CP's prediction sets with little data). I suggest a comparison is made with them (e.g., cross-CP and full CP) under the following setup:
- the proposal works as usual, by seeing both training and unlabelled sets
- the other two methods are only given the training data (i.e., no calibration data, for fairness).
Presumably, the proposal should perform better, because it has access to additional data. However, it is possible that the other two better exploit the information that is available in the training set.

3)
The fact that coverage guarantees are worse for larger $n$ and $m$ should be emphasized from the beginning.

4)
The term "probabilities" is confusing (and wrong); a more appropriate one is "confidence values".

>  the conventional approach can be seen as a particular case of Algorithm 2

This isn't correct. In Algorithm 2, the quantile is evaluated over a much larger multiset than the original algorithm, since it looks at all the possible labels.

**Ethical Concerns:**

["NO or VERY MINOR ethics concerns only"]

**Final Justification:**

Thanks for the rebuttal. I encourage the authors to include the cross-CP experiments in a revised version, along with an explanation on pros/cons.

**Limitations:**

yes

**Quality:**

3

**Strengths And Weaknesses:**

Strengths:
- new method that enables calibrating split CP on unlabelled points
- well-presented results
- good theoretical analysis

Weaknesses:
- it's hard to understand in practice what is the price to pay in the validity guarantee (excess coverage gap)
- no comparison against methods that try to achieve the same

---

> ### Author Rebuttal · Authors · 2025-07-30
>
> We would like to thank the Reviewer for his/her appreciation of the approach novelty and the theoretical analysis, and for the suggestions and comments provided. We are confident that the comments and questions raised are fully addressed below, and we would be happy to answer any further questions the Reviewer may have during the Authors/Reviewers discussion period.
>
> **Excess coverage gap with respect to conventional supervised split conformal prediction**
>
> As pointed out by the Reviewer, the difference in performance guarantees between the proposed methods and conventional supervised approaches is given by the term $G_{n,m}^F$ (excess coverage gap). As discussed qualitatively in Section 3.2 (Theorem 1) for general families of functions $\mathcal{F}$, such a term is small using broad families of functions and a sufficient number of samples. The result in Theorem 1 is applicable with wide generality, while more interpretable expressions of the excess coverage gap can be obtained for specific choices of family of functions. In particular, Theorem 2 presents a significantly more interpretable excess coverage gap for kernel-based families (equation (14)). In that case, the excess coverage gap corresponds to the optimum value achieved in a ridge-regression fit (with L1 loss) of the function 𝔛$_C$ using functions in a RKHS. As described on lines 275-284, the excess coverage gap decreases with the number of samples as $O(\sqrt{1/n +1/m})$ for any strictly positive definite kernel (e.g., Gaussian, Laplacian, and polynomial), with proportionality constant given by the minimum norm of functions that interpolate 𝔛$_C$ at the calibration instances. The magnitude of  such minimum norm depends on the smoothness of the function 𝔛$_C$ at the calibration instances. Such function may be highly non-smooth at the training instances, but is expected to be fairly smooth at the calibration instances for prediction rules that generalize.
>
> Notice also that the upper bounds for the excess coverage gap on lines 275-284 are computable in practice by solving a simple interpolation problem with closed-form solution, as described on lines 285-293. This fact allows to have a practical data-based method to select the kernel, as described in that paragraph and detailed on lines 2-7 of Algorithm 3.
>
> **Comparison with methods beyond split conformal prediction**
>
> The main contribution in the paper is a methodology that enables to carry out split conformal prediction without acquiring labels for the calibration samples. As described by the Reviewer, other methods for conformal prediction that are not based on split (or inductive) conformal prediction (e.g., cross-CP and full CP) do not rely on labeled calibration data as well, since they do not use calibration samples at all. However, split conformal prediction offers important intrinsic advantages with respect to other approaches such as cross-CP and full CP. In particular, split conformal prediction can be applied to any prediction model, including “black box” models with opaque or poorly understood training procedures, to produce valid predictive sets. On the other hand, methods such as cross-CP and full CP cannot be used with “black box” prediction rules and require to carry out multiple training stages in order to obtain the set-prediction rule. Full CP is highly inefficient and hardly implementable in practice since it requires to learn multiple prediction rules for each new test instance. Cross-CP is more efficient since it requires to carry out a fixed number of training stages (e.g., 5 or 10). However, Cross-CP does not provide performance guarantees for general types of prediction rules [R].
>
> Nevertheless, we agree with the Reviewer that it can be valuable to compare the proposed methods with alternative approaches beyond split conformal prediction. Therefore, we have carried out the experimental results suggested by the Reviewer using Cross-CP with 10-fold partitions (the implementation with full CP is not computationally feasible with strong prediction rules and sizable test sets). The following table shows the results achieved in "MNIST", "FashionMNIST" and "USPS"  datasets by Cross-CP and the proposed methods for unsupervised split conformal prediction. Specifically, the tables show the average coverage gap achieved using $m=1,000$ training samples and varying the number $n$ of unsupervised calibration samples.
>
> `MNIST’ dataset:
>
> |  | Cross-CP | Proposed (n=50) | Proposed (n=100) | Proposed (n=500) | Proposed (n=1000) | Proposed (n=2000) | Proposed (n=3000) |
> |--------------|----------|------------------|-------------------|-------------------|--------------------|--------------------|--------------------|
> |           Coverage Gap  | 0.011    | 0.033            | 0.024             | 0.014             | 0.014              | 0.012              | 0.011              |
>
>
> `FashionMNIST’ dataset:
>
> | | Cross-CP | Proposed (n=50) | Proposed (n=100) | Proposed (n=500) | Proposed (n=1000) | Proposed (n=2000) | Proposed (n=3000) |
> |--------------|----------|------------------|-------------------|-------------------|--------------------|--------------------|--------------------|
> |         Coverage Gap      | 0.013    | 0.027            | 0.022             | 0.016             | 0.014              | 0.013              | 0.012              |
>
>
> `USPS’ dataset:
>
> | | Cross-CP | Proposed (n=50) | Proposed (n=100) | Proposed (n=500) | Proposed (n=1000) | Proposed (n=2000) | Proposed (n=3000) |
> |--------------|----------|------------------|-------------------|-------------------|--------------------|--------------------|--------------------|
> |        Coverage Gap       | 0.012    | 0.033            | 0.024             | 0.014             | 0.013              | 0.012              | 0.012              |
>
> &nbsp;
>
> The new results show that, as the Reviewer correctly predicted, Cross-CP better exploits the information in the training samples and the proposed methods start to have better results than cross-CP for a sizable number of calibration samples. This behavior is completely sensible, both cross-CP and the proposed methods utilize the training samples, but the proposed methods use them only in a indirect manner to find probability distributions (weights) for calibration labels that match general expectations of training and calibration samples. On the other hand, cross-CP methods directly use the training samples to evaluate scores and determine quantiles. Therefore, cross-CP extracts more information from training samples but does not provide general performance guarantees, require multiple training stages, and cannot be used with “black-box” prediction rules. The indirect usage of training samples in the proposed methods allows to safely reuse these samples to obtain label probabilities (weights) for calibration samples even when they have been used to train the base classifier utilized.
>
> [R] Vovk, V. Cross-conformal predictors. Annals of Mathematics and Artificial Intelligence, 74(1), 9-28. 2015
>
> **Relation between coverage guarantees and number of samples $n$ and $m$**
>
> We do not really understand the comment from the Reviewer, since the coverage guarantees are indeed better for larger values of $n$ and $m$. In particular, the excess coverage gap decreases with n and m as $O(\sqrt{1/n+1/m})$ using kernel-based families of functions, as described on lines 275-284. For general families of functions, the excess coverage gap decreases with n and m with an order given by the Rademacher complexity of the family as described on lines 205-215.  In case we have misinterpreted the comment from the Reviewer, we would appreciate further guidance during the Author-Reviewers discussion period, so that we can completely clarify this aspect.
>
> **Change of terminology from “label probabilities” to “label weights”**
>
> We agree with the Reviewer that the usage of “label probabilities” for the solutions of optimization problem (4) can be somehow confusing since such terminology may indicate an unintended probabilistic interpretation. The term “confidence” suggested by the reviewer is better, but we feel it may lead to some different misleading interpretations. We think the term “label weights” is less prone to misinterpretation and we plan to use it in the final version of the paper. We would appreciate if the Reviewer let us know if such terminology is also appropriate in his/her view.
>
> **Conventional approach for supervised split conformal prediction as a particular case of Algorithm 2**
>
> As described by the Reviewer, the quantile obtained in the conventional approach for split conformal prediction is computed ranging only over the actual labels of calibration samples, while the quantile computed in the proposed methods ranges over all possible labels of calibration samples. The conventional approach in Algorithm 1 is a particular case of Algorithm 2 because taking in step 3 of Algorithm 2 the label probabilities (weights) corresponding to the actual calibration labels $p^*_i(y)=\mathbb{I}\lbrace y=Y_i\rbrace$, the quantile obtained by Algorithm 2 would be equal to that obtained by Algorithm 1. Notice that such label probabilities (weights) are 1 for the actual labels of calibration samples and 0 for the remaining labels so that the quantiles of Algorithm 1 and Algorithm 2 would coincide since the additional values computed for the quantile in Algorithm 2 would have zero weight.

---

> > ### Comment · Reviewer_cxnj · 2025-08-06
> >
> > Thank you for your detail responses, and for running these new experiments!
> >
> > > Excess coverage gap with respect to conventional supervised split conformal prediction
> >
> > Thanks for the explanation.
> >
> > > Comparison with methods beyond split conformal prediction
> >
> > These results are interesting. As a matter of fact, I would have predicted the opposite (i.e., your proposal to outperform them, due to additional info being available). Just to confirm: cross-CP was just given $n$, whereas your proposal was given both $n$ an $m$?
> > Agreed with your point on cross-CP being non-valid (not provably, at least), and with full-CP being too expensive.
> >
> > > Relation between coverage guarantees and number of samples $n$ and $m$
> > and
> > >Conventional approach for supervised split conformal prediction as a particular case of Algorithm 2
> >
> > Sorry, I stand corrected.
> >
> > > Change of terminology from “label probabilities” to “label weights”
> >
> > Label weights seems more factual (as long as you clarify in the text what you mean).

---

> > > ### Author Response · Authors · 2025-08-07
> > >
> > > We would like to thank the Reviewer for his/her appreciation of the paper. We are glad that our responses correctly addressed the Reviewer's comments.
> > >
> > > **Clarification regarding the new experimental results:**
> > >
> > > Yes, cross-CP was just given the training samples, while the proposed method was given both the training samples and unsupervised calibration samples. Notice that the paper uses $n$ to denote the number of calibration samples and $m$ to denote the number of training samples.
> > >
> > >
> > > The explanation for the worse performance for reduced values of $n$ (calibration samples) is as follows. Cross-CP directly uses the $m$ training samples to compute conformal quantiles. However, the method proposed uses the training samples only indirectly. Specifically, the training samples are used to obtain weights for the $n$ calibration samples; then, the conformal quantiles are computed using only the weighted calibration samples. For instance, in the new experimental results Cross-CP computes quantiles using $m=1,000$ samples, while the method presented computes quantiles using $n=50,100,500,...$ samples. Therefore, even if the methods presented were completely successful and the weights obtained using the training samples were perfect, in most of the columns of the table the proposed method computes quantiles with less samples than cross-CP, which explains the worse performance in those cases. The indirect usage of training samples in the proposed methods enables to achieve the intrinsic benefits of split conformal prediction even with unlabeled calibration samples, and also to provide performance guarantees.

---

### Official Review · Reviewer_85zz · 2025-07-01

**Clarity:** 3
**Significance:** 2
**Originality:** 2
**Rating:** 4
**Confidence:** 4

**Summary:**

This paper proposes a novel method to split conformal prediction for classification tasks, utilizing unsupervised calibration. The proposed method constructs set-prediction rules by integrating unsupervised calibration samples with supervised training samples previously employed to train the classification rule.

**Questions:**

- If supervised training samples are directly incorporated into the calibration set, can the coverage guarantee still be maintained?
- Can eq (4) ensure that the calibration and training samples become statistically indistinguishable? Could you provide a more detailed explanation to support this?
- Does eq (4) ensure the validity of the formula presented on line 168?
- The coverage on the Letter dataset appears consistently low; is this within an acceptable range?

**Ethical Concerns:**

["NO or VERY MINOR ethics concerns only"]

**Final Justification:**

The authors have provided detailed clarifications and additional experiments that satisfactorily address my main concerns. In particular, they explained the assumption of exchangeability, added results on ImageNet10 to strengthen empirical validation, and elaborated on the role of Eq. (4) and its implications for statistical equivalence. Therefore, I have decided to raise my score to 4.

**Limitations:**

Yes.

**Paper Formatting Concerns:**

No.

**Quality:**

2

**Strengths And Weaknesses:**

Strengths:
- This paper is well-written and easy to follow.
- To the best of my knowledge, this paper is the first to investigate the use of unsupervised calibration for split conformal prediction.

Weaknesses:
- The paper lacks a proof of exchangeability between the calibration set and the test set, which is a critical assumption for the validity of conformal prediction methods.
- The experimental validation is limited to small datasets; extending the evaluation to larger datasets, such as ImageNet, would significantly strengthen the robustness of the findings.

---

> ### Author Rebuttal · Authors · 2025-07-30
>
> We thank the Reviewer for the valuable suggestions and comments. We are confident that all points raised have been fully addressed below. Should the Reviewer feel that any aspect of the response requires further clarification, we would be happy to provide additional details during the authors/reviewers discussion period.
>
> **Exchangeability between the calibration set and the test set**
>
> We do not fully understand the comment from the Reviewer regarding a proof for exchangeability. As described on lines 67-68 the paper considers data samples (instance-label pairs) from the same random variable $(X,Y)$ that are drawn i.i.d. from the same data distribution. Therefore, the exchangeability between calibration and tests samples follows from the setting considered in the paper, which is the same as that in conventional split conformal prediction. We do not consider scenarios affected by distribution shifts in which the distribution of calibration samples is different to that of test samples (or that of training samples).
>
> In order to avoid any possible confusion, we will stress this fact in the preliminaries section by slightly modifying the first paragraphs as:
> "Let ... $\mathcal{D}$ be a dataset composed by i.i.d. copies of $(X,Y)$. The dataset is split into two groups .... All the data samples (for training, calibration, and test) are randomly drawn from the same data distribution, so that they are independent and, hence, exchangeable."
>
> In case we have misinterpreted the comment from the Reviewer, we would appreciate further guidance during the Author-Reviewers discussion period, so that we can completely clarify this aspect. We would also like to point out that an important contribution in the paper is that the theoretical analysis followed is novel (as briefly described on lines 164-174, and detailed in the proofs shown in the Appendices). Besides these technical novelties, the theoretical results presented also rely on the independence (exchangeability) of calibration and test samples (e.g., all the pairs $(X_i,Y_i), (\widetilde{X}_i,\widetilde{Y}_i), (X,Y)$ in the proofs are independently drawn from the same data distribution).
>
> **Additional experiments with ImageNet dataset**
>
> We agree with the Reviewer that the usage of ImageNet dataset in the experiments can further strengthen the experimental assessment, so that we have obtained new results with this dataset. In particular, we have utilized ImageNet10 composed by the first 10 classes of ImageNet because the methods proposed are not meant to be used in problems with a sizeable number of classes. This limitation is described on lines 220-224, 294-310, and 335-337, and will be further pointed out in a new paragraph “Limitations” describing the main two limitations of the methods proposed: the inadequacy to address problems with a significant number of classes, and the increase in complexity with respect to conventional split conformal prediction with supervised calibration.
>
> The results obtained with ImageNet10 dataset with a Resnet50 neural network are as follows:
>
> Row for Table 1 ($1-\alpha=0.9$)
>
> | Dataset    | Coverage Probability - Supervised | Coverage Probability - Unsupervised | Coverage Probability - Unsup. Naive | Prediction Set Size - Supervised | Prediction Set Size - Unsupervised |
> |------------|-----------------------------------|-------------------------------------|-------------------------------------|----------------------------------|-------------------------------------|
> | ImageNet10 | 0.9 (0.89, 0.91)                  | 0.89 (0.88, 0.90)                   | 0.86 (0.85, 0.87)                   | 1.66 (1.60, 1.71)                | 1.61 (1.55, 1.68)                   |
>
>
> Row for Table 3 ($1-\alpha=0.95$)
>
> | Dataset    | Coverage Probability (Supervised) | Coverage Probability (Unsupervised) | Coverage Probability (Unsup. Naive) | Prediction Set Size (Supervised) | Prediction Set Size (Unsupervised) |
> |------------|-----------------------------------|--------------------------------------|-------------------------------------|-----------------------------------|--------------------------------------|
> | ImageNet10 | 0.95 (0.94, 0.96)                 | 0.94 (0.93, 0.95)                    | 0.93 (0.92, 0.93)                   | 2.22 (2.13, 2.29)                 | 2.05 (1.98, 2.13)                    |
>
>
> Row for Table 4 ($1-\alpha=0.85$)
>
> | Dataset    | Coverage Probability (Supervised) | Coverage Probability (Unsupervised) | Coverage Probability (Unsup. Naive) | Prediction Set Size (Supervised) | Prediction Set Size (Unsupervised) |
> |------------|-----------------------------------|--------------------------------------|-------------------------------------|-----------------------------------|--------------------------------------|
> | ImageNet10 | 0.85 (0.85, 0.86)                 | 0.85 (0.84, 0.86)                    | 0.80 (0.79, 0.81)                   | 1.37 (1.33, 1.41)                 | 1.37 (1.32, 1.41)                    |
>
> &nbsp;
>
> These new results show that the methods proposed are also effective in the dataset ImageNet10. As in the other experimental results in the paper, the performance of the methods proposed is only slightly inferior to that of conventional methods with supervised calibration.
>
> Overall, the paper shows experimental results with 8 datasets (9 counting ImageNet10) with number of samples ranging from 6k to 581k (ImageNet10 has 13k samples), number of classes ranging from 6 to 26, and input dimensionality ranging from 16 to 150k (ImageNet10 is composed by 3x224x224 images).
>
> **Coverage guarantees if training samples are incorporated in the calibration set**
>
> The performance guarantees presented in the paper rely on the independence among the samples used for training and calibration (stated on lines 67-68). For instance, the bounds in (33), (34), (36), and (38) (among others) are obtained using such independence. As described in a previous response, we will further stress the independence among samples at the end of the second paragraph of the preliminaries section. Nevertheless, the Reviewer raises a quite interesting and subtle point. If supervised training samples are directly incorporated into the calibration set, performance bounds somehow similar (in form) to those in the paper could be obtained. However, such type of approaches would not be effective in practice and the quantitative values of the bounds would be much higher, at least using common modern classifiers that often fit almost perfectly the training samples. As discussed on lines 280-284, the performance guarantees of the methods proposed rely on the smoothness at the calibration instances of the function 𝔛$_C$ (obtained from the base classifier score). Such a function is often highly non-smooth at the training instances (e.g., using classifiers that perfectly interpolate the training samples). On the other hand, in common cases the function 𝔛$_C$ is fairly smooth at calibration instances that are randomly drawn independently from the training instances. Notice that cases in which the classifier score is highly non-smooth for samples independent to those in the training set would correspond to highly poor classifiers. In the final version of the paper, we will expand the paragraph on lines 280-284 to discuss these aspects.
>
> **Optimization in (4), statistical equivalence of calibration and training samples, and formula on line 168**
>
> Solutions of optimization problem (4) correspond to training and calibration sets that are statistically indistinguishable by the corresponding IPM-based two-sample test. The optimal value of (4) is not larger than the value of the IPM test achieved using the labeled calibration and training samples $\lbrace(X_i,Y_i)\rbrace$ and $\lbrace(\widetilde{X}_i,\widetilde{Y}_i)\rbrace$ because the value of such IPM test coincides with the value of (4) for the specific label probabilities $p_i^*(y)=\mathbb{I}\lbrace y=Y_i\rbrace$. Then, an IPM-based test would not find a significant statistical difference between training and calibration sets corresponding to solutions of (4) since the value of the test statistic would be not larger than that achieved by two sets that are composed by independent samples from the same distribution.
>
> We would also like to point out that we see such statistical equivalence as an insightful way to illustrate the methodology presented. The technical proofs of Theorem 1 and Theorem 2 in Appendices B and C are not consequences of such statistical equivalence. For instance, Theorem 1 is valid also in cases where the label probabilities $p_i^*(y)=\mathbb{I}\lbrace y=Y_i\rbrace$ do not correspond to a feasible solution for (4).
>
> Label probabilities found solving optimization in (4) ensure the validity of the formula on line 168 due to a similar rationale as above. Specifically, the objective value in (4) is $O(\sqrt{1/n+1/m})$ for the label probabilities that correspond to the actual calibration labels  because samples $\lbrace (X_i,Y_i)\rbrace$, $\lbrace (\widetilde{X}_i,\widetilde{Y}_i)\rbrace$ are i.i.d. Therefore, the optimal value of (4) is small for a sufficient number of calibration and training samples, leading to the approximation on line 168.
>
> **Coverage gap in "Letter" dataset and in other cases with a sizeable number of classes**
>
> As described above (and on lines 220-224, 294-310), the proposed methods provide poor performance in cases with a sizeable number of classes ("letter" dataset has 26 classes). The poor coverage achieved in "letter" is specifically accounted in lines 335-337 “In accordance with the discussion in Section 3.2 regarding the weakened guarantees in scenarios with many classes, "Letter" –which includes 26 classes– is the only dataset showing significantly poorer performance.” As described in previous responses, the final version of the paper will include a specific paragraph dedicated to state the limitations of the proposed methods in terms of the number of classes.

---

> > ### Comment · Reviewer_85zz · 2025-08-05
> >
> > Most of my concerns have been addressed. I have increased my score to 4.

---

> ### Author Response · Authors · 2025-08-05
>
> We would like to thank the Reviewer for his/her appreciation of the paper and for raising the score based on our detailed responses. If the Reviewer has any remaining questions or concerns, we would be glad to provide further clarification before the end of the Auhtors/Reviewers discussion period on August 8.

---

### Official Review · Reviewer_Nrn4 · 2025-07-02

**Clarity:** 3
**Significance:** 2
**Originality:** 2
**Rating:** 4
**Confidence:** 4

**Summary:**

This paper proposes a novel method for conformal prediction using unsupervised calibration data. The authors introduce a technique for assigning label probabilities to each calibration example by leveraging the labeled training data. These probabilities are obtained by solving an optimization problem that minimizes the integral probability metric between the distribution of labeled training samples and the pseudo-labeled calibration data.

**Questions:**

1-	Could the authors provide experimental results on runtime to compare the proposed method with supervised and naive unsupervised approaches?

2-	Theorem 2 states that the guarantees hold for any data distribution. Does this imply that the proposed algorithm remains valid even when the exchangeability assumption between test and calibration data is violated?

3-	How does the proposed method compare to existing weakly-supervised conformal prediction methods in terms of coverage and prediction set size?

4-	As noted in the paper, achieving a small coverage gap requires the function class $\mathcal{F}$ to accurately approximate the indicator function $\mathcal{X}_C$. Could the authors elaborate on how the choice of $\mathcal{F}$ affects the prediction set size in practice

**Ethical Concerns:**

["NO or VERY MINOR ethics concerns only"]

**Final Justification:**

The paper provides a solid theoretical foundation for the proposed method. During the rebuttal, the authors addressed my concerns; therefore, I have decided to increase my confidence in the score.

**Limitations:**

The proposed method may not scale well to settings with a large number of labels. Its performance relies on the chosen function class $\mathcal{F}$ being able to accurately approximate the function $\mathcal{X}$,

**Quality:**

3

**Strengths And Weaknesses:**

Strengths: The paper provides a solid theoretical foundation for the proposed method, including performance guarantees and computational complexity analysis. It is also well-written and clearly presented.

Weakness: Please check the questions.

---

> ### Author Rebuttal · Authors · 2025-07-30
>
> We would like to thank the Reviewer for his/her appreciation of the solid theoretical foundation presented, and for the suggestions and comments provided. We are confident that the comments and questions raised are fully addressed below, and we would be happy to answer any further questions the Reviewer may have during the authors/reviewers discussion period.
>
> **Runtime of the methods proposed with respect to conventional split conformal prediction**
>
> Appendix E.2 in the supplementary materials experimentally assess the runtime of the methods proposed (lines 294-310 in Section 4 describe the theoretical computational complexity). We plan to use the additional space allowed in the final version to incorporate in the main paper the running times in the experimental results. Figure 3 in Appendix E.2 shows the running times of the methods presented as the number of calibration samples increases in 3 datasets with different number of classes (7 in "Drybean", 10 in "CIFAR10", and 26 in "Letter"). As shown in the figure, the presented methods have running times ranging from tens to hundreds of seconds for cases with up to 10 classes and up to 3,000 calibration samples. In line with the discussion in Section 4 of the paper regarding the computational complexity of the methods presented, the only problematic cases are those composed by a sizable number of classes, since the computational complexity of the methods presented increases cubically with the number of classes. On the other hand, the conventional approaches for conformal prediction (supervised and naive) have much smaller running times (less than a second) since they only require to computed one quantile.
>
> The significant difference in complexity and the unsuitability for cases with many classes are the main limitations of the methods proposed, as pointed out on lines 47-48, 220-224, 294-310, and 335-337. The increase in complexity does not limit the practicality of the methods proposed. In particular, the running time of the methods presented is significantly lower than that of usual training stages. In addition, it can be argued that a running time as low as that of the supervised approach is not needed in practice since the calibration stage is commonly carried out only once. These aspects will be further discussed in the final version of the paper. In particular, we will add a new paragraph “Limitations” dedicated to state the main limitations of the methods proposed.
>
> **Exchangeability between test and calibration data**
>
> In the paper, all data samples (instance-label pairs) are described by the same random variable $(X,Y)$ (see lines 66-69) so that they are samples from the same distribution, that is, the paper does not address scenarios affected by distribution shifts. The theoretical guarantees of the methods proposed (as those in conventional supervised split conformal prediction, see line 87) are valid for any data distribution, that is, for any distribution of the random variable $(X,Y)$ describing instance-label pairs of training, calibration, and test samples. As in the conventional approach, this generality does not affect the exchangeability of calibration and test samples, it just describes that the results are valid for an arbitrary data distribution (of both calibration and test samples).
>
> **Comparison with existing methods for weakly-supervised conformal prediction**
>
> To the best of our knowledge, the paper presents the first method for conformal prediction that can utilize unsupervised calibration samples. Existing methods for weakly-supervised conformal prediction cannot be used with unlabeled calibration samples. Specifically, the methods in [5] and [7] require to have fully labeled calibration samples (the labels are allowed to be affected by random noise). In the methods [6] and [8], some of the calibration samples can be unlabeled (missing labels) but a significant proportion of the calibration samples need to be accurately labeled. Therefore, it is not really possible to carry out a fair comparison between the proposed methods and existing methods for weakly-supervised conformal prediction.
>
> The comparisons carried out in the paper are with respect to the benchmark approach of supervised conformal prediction that describes the best possible performance and with respect to a naive approach devised by us to have a reference for the performance with a simple approach. The results presented show that the proposed methods can achieve performances near to the benchmark of supervised calibration and significantly better than the naive approach.
>
> **Relation between the choice of the function class and prediction set sizes**
>
> The choice of the function class can directly affect the coverage gap (as described in Theorems 1 and 2). However, its influence on the prediction set size is only indirect. Specifically, cases with a negative coverage gap (i.e., the actual coverage is smaller than the target coverage) result in somehow smaller prediction set sizes since the conformal quantile obtained in those cases is smaller than the ($1-\alpha$)-quantile of the distribution. Reciprocally, cases with a positive coverage gap results in somehow bigger prediction set sizes. These facts can be observed in the experimental results, for instance the coverage gap in Figure 1 with unsupervised calibration tends to be slightly higher than that with supervised calibration (left panel), and the prediction set size with unsupervised calibration tends to be slightly higher (right panel). Similar behavior can be observed in other results like in Table 1.
>
> The choice of the function class affects the coverage gap, but such a gap is small for a large-enough number of samples. In particular, the usage of too small or too complex function classes would result in larger coverage gaps (see discussion for the bias-variance trade-off for the choice of the function class on lines 205-215). As discussed on lines 275-284, the usage of function classes defined by positive definite kernels (e.g., Gaussian, Laplacian, and polynomial) results in coverage gaps that decrease with the number of samples as $O(\sqrt{1/n+1/m})$, with smaller proportionality constant in cases where the kernel used is well-adapted to the dataset at hand. As shown on lines 285-293 and lines 2-7 in Algorithm 3, the proposed methods enable to effectively select the kernel used so that the corresponding coverage gap is small (as corroborated in the experimental results).

---

> > ### Comment · Reviewer_Nrn4 · 2025-08-05
> >
> > Thank you to the authors for the detailed responses. I want to maintain my current score and have increased my confidence in it.

---

> > > ### Author Response · Authors · 2025-08-05
> > >
> > > We thank the Reviewer for recognizing the novelty and theoretical rigor of results presented, and the level of detail in our responses.

---

### Official Review · Reviewer_bFNU · 2025-07-03

**Clarity:** 3
**Significance:** 3
**Originality:** 3
**Rating:** 5
**Confidence:** 3

**Summary:**

This paper considers the problem of obtaining a conformal prediction set where the conformal score thresholds are calibrated using an unlabeled calibration set. The proposed method learns class probabilities by matching the joint distribution between the labeled training set (X̃, Ỹ) and the pseudo-labeled calibration set $(X = x, Y = j)$, where each is weighted by its estimated probability $P(Y = j | X = x)$, measured using an integral probability metric (IPM). The paper then shows that, under mild assumptions on the function space used for distribution matching and on the relationship between this space and the conformal set indicator function, the proposed approach can provide bounds on the coverage rate.

**Questions:**

As discussed above, a central claim of the paper is that the proposed class-probability optimization does not require a highly accurate base classifier trained on the labeled data. However, the constraint involving the base classifier predictions (specifically the bound on Bp) appears to implicitly require that the learned probabilities align reasonably well with those from a classifier. Could the authors clarify: how accurate does the base classifier need to be in practice for the method to perform well? Is there a failure mode if the base classifier is poorly calibrated or highly inaccurate?

In addition, it would be helpful to include a more practical comparison with existing approaches for conformal prediction using unlabeled calibration data, as well as standard classifier calibration methods (e.g., temperature scaling, Dirichlet calibration, use splits of the labelled training set to calibrate). This would help contextualize the proposed method’s strengths and limitations in applied settings.

**Ethical Concerns:**

["NO or VERY MINOR ethics concerns only"]

**Final Justification:**

After discussing with the authors, I would like change my position to be (5: Accept). In general it is sounding approach for the domain of conformal prediction. If there is anything, I would suggest to investigate further the links to related works as in PPI and post-hoc calibration, as well as added some case studies on the impact of the base predictive models.

**Limitations:**

Yes

**Quality:**

3

**Strengths And Weaknesses:**

(+) The proposed approach is sound and, to the best of the reviewer’s knowledge, novel. This is the first work to leverage the idea of joint distribution matching on an unlabeled calibration set to construct conformal prediction sets. The theoretical framework is compelling and naturally connects to the literature on two-sample testing, particularly via maximum mean discrepancy (MMD) (Gretton et al., 2012). The authors also extend their method to a kernel-based version, making this connection explicit and providing a practical implementation pathway.



(–) That said, a key weakness of the paper lies in the lack of discussion of related work and practical guidance on method selection. Specifically:

* Conformal prediction with unlabeled calibration data has been studied in prior work [1,2,3], A detailed comparison of assumptions, theoretical guarantees, and empirical performance would strengthen the paper:




* Prediction-powered inference (PPI) [4] is a related framework where unlabeled data is paired with predicted probabilities to perform valid inference. While this paper does not directly use hard classifier outputs, the optimization still implicitly assumes reasonable prediction quality, creating conceptual overlap worth acknowledging.

* The literature on classifier probability calibration [5, 6] focuses on adjusting individual predictive confidence. While such methods lack coverage guarantees like conformal prediction, they offer fine-grained, per-instance calibration that may be practically useful in similar contexts. A clearer discussion of the tradeoffs could help position this work.


(1) Zhou et al., SemiCP: Semi-Supervised Conformal Prediction With Unlabeled Nonconformity Score

(2) Han et al., Conformal Calibration under Distribution Shift

(3) Lienen et al., Conformal Prediction with Weak or Credal Labels

(4) Gibbs & Jordan, Prediction-powered inference

(5) Guo et al. (2017), On Calibration of Modern Neural Networks

(6) ull et al., Beyond Temperature Scaling: Obtaining Well-Calibrated Multi-Class Probabilities with Dirichlet Calibration

---

> ### Author Rebuttal · Authors · 2025-07-30
>
> We would like to thank the Reviewer for his/her appreciation of the paper and for the suggestions and comments provided. We are confident that the comments and questions raised are fully addressed below, and we would be happy to answer any further questions the Reviewer may have during the authors/reviewers discussion period.
>
> **Prior work on conformal prediction with unlabeled calibration data**
>
> To the best of our knowledge, there are not previous works on conformal prediction with unlabeled calibration data. Regarding the papers [1,2,3] suggested by the reviewer: the semi-supervised approach in [1] requires both labeled and unlabeled calibration samples, while the presented method only uses unlabeled calibration samples. In addition, such a paper is currently under review and was submitted to arxiv 12 days after the full paper submission deadline for this conference, so it was not possible for us to discuss such a work. In the final version of the paper, we will use the extra space allowed to include a discussion of the semi-supervised approach in the description of methods for split conformal prediction with weakly supervised calibration (currently lines 35-41). Regarding references [2] and [3] suggested by the Reviewer, we have not been able to find them, so it is not clear for us what prior work the Reviewer is referring to.
>
>
> **Relation with prediction-powered inference and calibration methods**
>
> As described by the Reviewer, there is some conceptual overlap with the methods proposed in the paper and methods for prediction-powered inference and calibration. Such relation will be discussed in the final version of the paper together with the relevant references.
>
> The naive approach used as a baseline for split conformal prediction can be seen as an application of prediction-powered inference where a classification model is used to obtain labels for the calibration samples. As described in the paper, such type of approaches have inferior performance than the proposed methods (lines 111-115, 225-232, and 337-339). The proposed approach, can be thought of as obtaining "pseudolabels" for the calibration samples in terms of probabilities. However, the proposed methods do not aim to maximize the number of "pseudolabels" that agree with actual labels but to find "pseudolabels" that enable to estimate general expectations (and hence quantiles) similarly as the actual labels.
>
> Calibration methods are related to conformal prediction techniques since both frameworks aim to better account for the predictive uncertainty of learning methods. However, calibration and conformal prediction methods account for such uncertainty in rather different ways. Conformal prediction aims to obtain set-predictions that contain the true label with high probability while calibration methods aim to obtain probabilistic predictions that are consistent (calibrated) with the actual frequency of different labels. Therefore a quantitative comparison of both types of approaches is hardly possible.
>
> **Relation between the accuracy of the base classifier and the performance achieved by the proposed methods**
>
> The accuracy of the base classifier is closely related to the performance of the naive approach used as baseline, but not to the performance of the proposed methods. In particular, the coverage gap for the naive approach is of the order of the base classifier error, while the coverage gap for the methods proposed decreases as $O(\sqrt{1/n+1/m})$ regardless of the error of the base classifier (see also lines 111-115, lines 225-232, and lines 275-277). These theoretical results are also in agreement with the experimental results, which demonstrate that the naive approach only provides acceptable performance in cases where the base classifier has very low error ("CIFAR10" and  "USPS" datasets). On the other hand, the proposed methods obtain a performance very close to that with supervised calibration samples in all the experimental results. Specifically, the error of the base classifier used in each dataset is as shown in the following table
>
> | Dataset             | Drybean | Forestcov | Satellite | USPS | MNIST | Fashion | CIFAR10 | Letter |
> |---------------------|---------|-----------|-----------|------|--------|---------|---------|--------|
> | Base classifier error | 0.12    | 0.33      | 0.12      | 0.04 | 0.06   | 0.18    | 0.04    | 0.34   |
>
>
> Such errors are significantly higher than the coverage gap of the proposed methods.
>
> The constraint involving the base classifier predictions ($\mathbf{B}\mathbf{p} \preceq \mathbf{b}$) does not imply that the probabilities obtained by solving optimization (4) (or (12)) align with those provided by the base classifier. Such a constraint implies that expected losses of the base classifier, estimated using the optimized label probabilities, are not larger than an upper bound on that expected loss. For instance, if the upper bound for the expected loss is obtained as
>
> $$
> \mathbf{b} = \frac{1}{m} \sum_{i \in [m]} \ell(h(\widetilde{X}_i), \widetilde{Y}_i) + S
> $$
> for $S$ being the sample standard deviation of the above sample mean, then the constraint ($\mathbf{B}\mathbf{p} \preceq \mathbf{b}$) becomes
>
> $$
> \sum_{i \in [n]\, y \in \mathcal{Y}} p_i(y) \ell(h(X_i), y) \leq \frac{1}{m} \sum_{i \in [m]} \ell(h(\widetilde{X}_i), \widetilde{Y}_i) + S
> $$
> Such a constraint is not directly related to the alignment between the outputs of the base classifier ($h(X_i)$) and the optimized label probabilities ($p_i(y)$). Instead, the constraint is related to how well the probabilities $\lbrace p_i(y)\rbrace$ align with the empirical distribution of the labeled training samples, in terms of matching expected values of the function $g(X,Y) = \ell(h(X), Y)$. An expectation estimate of any function $g(X,Y)$ could be used in optimization problem (4); the use of expected loss estimates for the base classifier is a convenient choice because most methods return such estimates as part of the training process.

---

> > ### Comment · Reviewer_bFNU · 2025-08-04
> >
> > Thanks for the discussion, overall with the proposed modifications I think this paper could be a nice addition to the conference, a few further comment below:
> >
> > 1: Apologies for the inaccurate references, the reviewer did use a LLM to proof read the raw reviews and didn't check the final results carefully. Here are the links for the two references:
> >
> > https://www.stat.berkeley.edu/~ryantibs/statlearn-s23/lectures/conformal_ds.pdf (should be very close to the context of this paper as the assumption is $P(Y|X)$ remains the same and the question is to adjust according to the shifted $P(X)$ when there is no labels).
> >
> > https://proceedings.mlr.press/v204/lienen23a/lienen23a.pdf (draw conditional distributions on unlabelled data and perform conformal predictions).
> >
> > 2: The discussion on the links to PPI is a nice addition and would be great to see them in the final paper. Regarding post-hoc calibration, the reviewer agrees that it is non-trivial to directly compare the two due to the different objectives, but some simple checks on the coverage levels would be useful to provide some intuition to the readers (maybe in future work).
> >
> > 3: Thanks for the elaboration. In the extreme case of a constant classifier that predict the marginal label distribution (i.e. $P(Y|X=x) = P(Y) \forall x$), will the proposed approach be able to recover the information on $x$ even it is not given in the raw classifier?

---

> > > ### Author Response · Authors · 2025-08-04
> > >
> > > We thank the Reviewer for his/her appreciation and support for the paper. We also thank the Reviewer for clarifying the references for conformal prediction and for the suggestions provided.
> > >
> > > Both references are related to the paper but aim to address different settings/problems:
> > >
> > > -The work in “Conformal Prediction Under Distribution Shift” addresses situations affected by distribution shifts, where the probability distribution of calibration instances (features) is different to that at test. The paper is related to our work in the sense that it provides conformal prediction methods that go beyond the conventional approach, but the setting addressed in the submitted paper is different. In particular,  all data samples (instance-label pairs) in the paper are described by the same random variable (see lines 66-69) so that they are samples from the same distribution, that is, the submitted paper does not address scenarios affected by distribution shifts.
> > >
> > > -The work in "Conformal Credal Self-Supervised Learning" utilizes conformal prediction methods to improve pseudolabeling of unlabeled samples so that they can be better used for learning (not for conformal prediction). The paper is related to our work in the sense that it addresses situations with unlabeled samples. However, the work in "Conformal Credal Self-Supervised Learning" does not aim to propose new methods for conformal prediction, and utilizes the conventional approach for conformal prediction to get pseudolables.
> > >
> > > Regarding the question for a situation with a constant classifier: the usage of conformal prediction with that type of classifier would result in a correct coverage but uninteresting prediction sets. In particular, the prediction sets would be also constant; for any instance the prediction set would be composed by the labels for which P(Y) is smaller than the conformal quantile. These uninteresting behavior would arise both using the proposed methods and existing approaches for conformal prediction. In general, the performance of conformal prediction methods is poor when the scores provided by the base classifier are poor. The methods proposed are not more strongly affected than conventional approaches by the accuracy of the base classifier. For instance, as described in the responses above, the proposed methods maintain a performance very close to that achieved with supervised calibration even using inaccurate base classifiers.

---

### Comment · Area_Chair_Kggo · 2025-08-04

Dear Reviewers,

Please review the authors' rebuttal and update your evaluation as needed.

AC

---

### Decision · Program_Chairs · 2025-09-17

**Decision:**

Accept (poster)

**Comment:**

This paper introduces a novel and theoretically sound method for split conformal prediction using unsupervised calibration, a significant departure from standard approaches that require labeled data. The core contribution is leveraging joint distribution matching to learn classification rule thresholds, which reviewers found to be a compelling and well-grounded idea. Initially, reviewers raised valid concerns regarding the empirical evaluation on larger datasets, the comparison to related work like prediction-powered inference, and practical aspects such as runtime. The authors provided a thorough and convincing rebuttal. They included new experimental results on ImageNet10, which strengthened the paper's empirical claims. Furthermore, they clarified the theoretical assumptions and contextualized their work effectively against existing literature, satisfying the reviewers' questions. The productive discussion period solidified the reviewers' confidence in the work, leading to a consensus that this is a valuable contribution.